# Use What You Know: Causal Foundation Models with Partial Graphs

**Arik Reuter** [1 2]  **Anish Dhir** [3]  **Cristiana Diaconu** [1]  **Jake Robertson** [4 5]  **Ole Ossen** [5]  **Frank Hutter** [4 6 5]
**Adrian Weller** [1 7]  **Mark van der Wilk** [8]  **Bernhard Schölkopf** [2 6]

## Abstract

Estimating causal quantities traditionally relies on bespoke estimators tailored to specific assumptions. Recently proposed Causal Foundation Models (CFMs) promise a more unified approach by amortising causal discovery and inference in a single step. However, in their current state, they do not allow for the incorporation of any domain knowledge, which can lead to suboptimal predictions. We bridge this gap by introducing methods to condition CFMs on causal information, such as the causal graph or more readily available ancestral information. When access to complete causal graph information is too strict a requirement, our approach also effectively leverages partial causal information. We systematically evaluate conditioning strategies and find that injecting learnable biases into the attention mechanism, together with a graph-convolutional encoder, is a highly effective method to utilise full and partial causal information. Our experiments show that this conditioning allows a general-purpose CFM to match the performance of specialised models trained on specific causal structures. Overall, our approach addresses a central hurdle on the path towards all-in-one causal foundation models: the capability to answer causal queries in a data-driven manner while effectively leveraging any amount of domain expertise.

## 1. Introduction

Causal questions are central to science and decision-making. Traditional approaches to answer those questions (Imbens & Rubin, 2015; Pearl, 2009) are tailored to specific settings where certain assumptions hold. This effectively makes estimating causal effects a two-step process: **(i)** make identification assumptions (e.g. through a causal graph), **(ii)** choose an estimator based on those assumptions. This approach has produced a sizeable toolbox of estimators, and selecting amongst them requires substantial expertise. In other areas of machine learning, a similar reliance on human expertise has been reduced by using foundation models that are trained once on diverse datasets and adapt their predictions *in context* to the specific task at hand, eliminating the need for bespoke estimators (Wang et al., 2024; Bodnar et al., 2025; Brown et al., 2020).

Recently, Prior-Data-Fitted Networks (PFNs; Hollmann et al. 2022) have been proposed as a foundation model paradigm for tabular data, achieving state-of-the-art performance on tabular benchmarks (Erickson et al., 2025), without relying on human experience to tailor a method to a specific setting. Motivated by this, several PFN-based foundation models for causal inference have been proposed (Robertson et al., 2025; Dhir et al., 2025b; Ma et al., 2025; Sauter et al., 2025). These Causal Foundation Models (CFM) are trained to predict causal effects on numerous synthetic datasets generated from known causal models. With the training data specifying the prior, these models implicitly integrate over a posterior on causal graphs and functions to provide the causal effect for a test dataset (Dhir et al., 2025b; Robertson et al., 2025). This allows for estimating causal effects by training a CFM, either with causal assumptions reflected in the training data (by restricting the causal graph), or with minimal causal assumptions (by including all possible causal graphs), while capturing uncertainty from finite samples and structural ambiguity in both cases through the Bayes posterior.

However, when partial domain knowledge is available, restricting the training data (prior) to reflect the domain knowledge requires expensive retraining of CFMs. On the other hand, marginalising over *all* structures can yield unnecessarily conservative estimates when partial domain knowledge

[1]University of Cambridge, Cambridge, United Kingdom [2]Max Planck Institute for Intelligent Systems, Tübingen, Germany [3]Gatsby Computational Neuroscience Unit, University College London [4]Prior Labs, Freiburg, Germany [5]University of Freiburg, Freiburg, Germany [6]ELLIS Institute Tübingen, Tübingen, Germany [7]The Alan Turing Institute, London, United Kingdom [8]University of Oxford, Oxford, United Kingdom. Correspondence to: Arik Reuter <ar2364@cam.ac.uk>.

*Proceedings of the 43rd International Conference on Machine Learning*, Seoul, South Korea. PMLR 306, 2026. Copyright 2026 by the author(s).

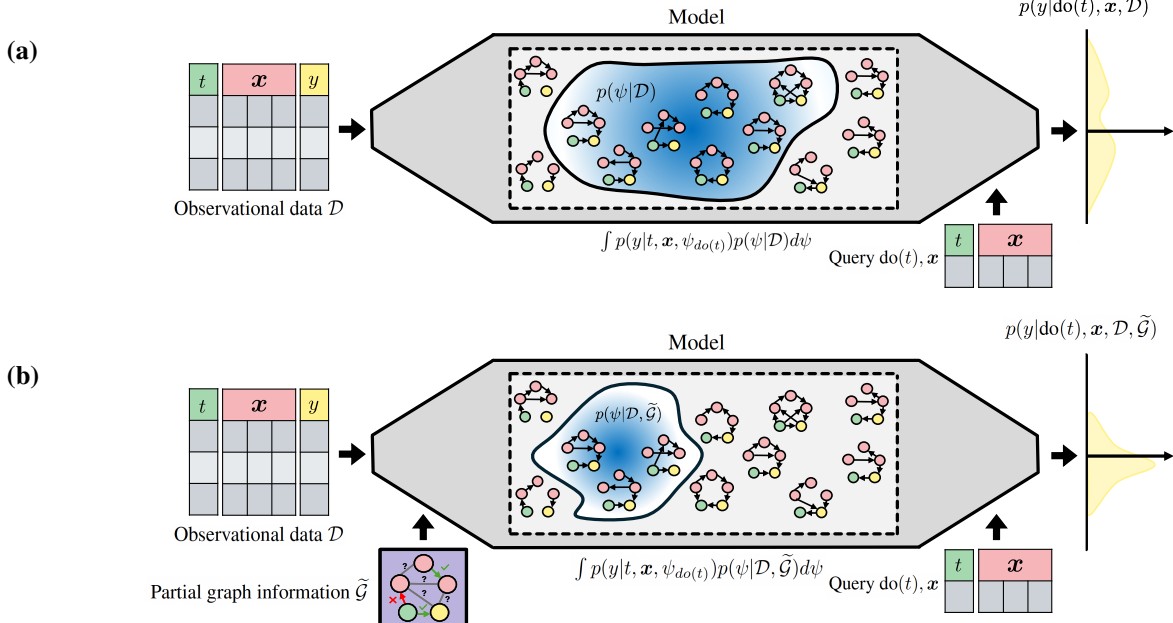

*Figure 1.* Causal Foundation Models based on PFNs for predicting the effect of causal interventions: A model takes as input observational data $\mathcal{D}$ and a query-point comprising an intervention $\text{do}(t)$ together with a feature vector $\boldsymbol{x}$. The model, implemented as a transformer, outputs the posterior of the causal effect $p(y|\text{do}(t), \boldsymbol{x}, \mathcal{D})$. This can be seen as the model implicitly marginalising over causal structures (SCMs) $\psi$, based on the posterior probability $p(\psi|\mathcal{D})$ that $\psi$ could have generated the observational data $\mathcal{D}$. **(a)** The model can only leverage observational data to compute its posterior belief $p(\psi|\mathcal{D})$ over possible causal structures (visualized as the region shaded in blue). When using this posterior to compute the posterior predictive $p(y|\text{do}(t), \boldsymbol{x}, \mathcal{D}) = \int p(y|t, \boldsymbol{x}, \psi_{\text{do}(t)})p(\psi|\mathcal{D})d\psi$, this leads to high uncertainty, and possibly imprecise predictions. **(b)** In contrast, providing the model with partial graph information $\widetilde{\mathcal{G}}$ allows to narrow down the set of causal structures that have high posterior mass $p(\psi|\mathcal{D}, \widetilde{\mathcal{G}})$. This yields a more concentrated posterior predictive $p(y|\text{do}(t), \boldsymbol{x}, \mathcal{D}, \widetilde{\mathcal{G}})$, and thus more accurate predictions of the outcome.

is available, leading to imprecise predictions by assigning weight to causal worlds that experts can rule out (see Figure 1). We thus identify a central shortcoming: *Existing CFMs cannot condition on prior causal information when performing causal inference at test time.*

In this work, we study techniques for flexibly conditioning CFMs on causal information. To allow for conditioning on causal information that might be more readily available than the exact causal graph, we focus on conditioning on *causal ancestral* information. Even so, access to all the ancestral information can be unrealistic in certain applications. Hence, we focus on three regimes: **(a)** there is ancestral knowledge about all the causal variables, **(b)** there is partial ancestral causal knowledge among a few of the variables (which can be represented as a partial ancestral graph), **and (c)** there is no causal knowledge between variables. Our proposed method allows for expressing all three cases above in a *single* model, allowing CFMs to condition on not only observational data in-context, but also on any amount of available domain knowledge (or lack thereof). In particular, we address the following three questions:

**1.) What type of causal information should one condition on in CFMs?** We argue that *partial ancestral information* provides a natural, flexible, and practically useful representation of causal domain knowledge, subsuming settings with no, partial, or complete causal information.

**2.) How to condition on causal information?** We systematically study different ways of incorporating partial graph information into CFMs and empirically show that *learnable attention biases*, together with a graph-convolutional encoder, are an effective and robust mechanism to enable the use of partial ancestral information in CFMs.

**3.) Does this translate to complex data?** We evaluate the proposed mechanisms for incorporating partial graph information into CFMs on a complex and realistic simulator of causal structures based on priors for existing Tabular Foundation Models, and on established semi-synthetic causal benchmarks. We find that ancestry-conditioning, even when it is just partially known, yields substantial improvements.[1]

---

[1] The code for our natively causal prior, model training, and our experiments is available at https://github.com/ArikReuter/Graphs4CausalFoundationModels.

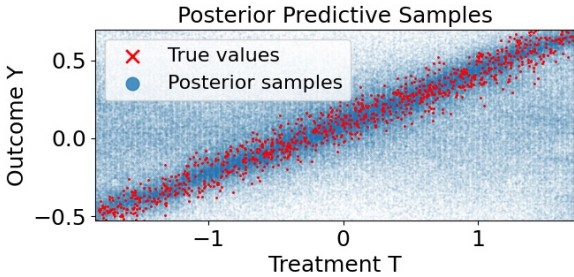 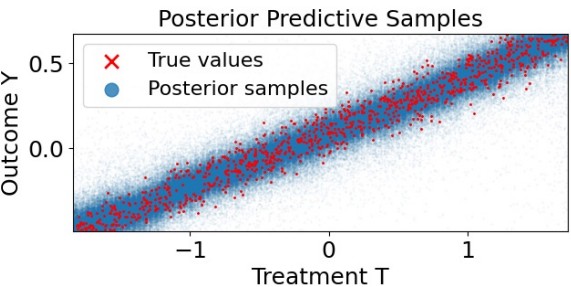

*Figure 2.* Posterior predictive samples in the two-node case where $T$ causes $Y$ ($T \rightarrow Y$); more specifically treatment and outcome are related via $Y = 0.25 \cdot T + \epsilon$, for $\epsilon \sim \mathcal{N}(0, 0.1)$. Here, the causal direction is not identifiable from observational data. **Left**: Without telling the model $Q_\theta$ that is trained on arbitrary causal structures (as, e.g., in Robertson et al. (2025) or Dhir et al. (2025b)) the correct causal direction, it outputs a mixture distribution between the posteriors for the two causal directions $Q_\theta(Y|\text{do}(T)) = 0.5 \cdot P(Y|\text{do}(T), Y \rightarrow T) + 0.5 \cdot P(Y|\text{do}(T), T \rightarrow Y)$. **Right**: When conditioned on the right causal direction, however, the model's output aligns with the correct causal effect: $Q_\theta(Y|\text{do}(T), T \rightarrow Y) = P(Y|\text{do}(T), T \rightarrow Y)$, leading to more precise predictions.

## 2. Background

The aim of this paper is to investigate the incorporation of domain knowledge into Causal Foundation Models to improve their predictions of causal effects. We review the necessary background regarding causal structures, interventions, interventional distributions, and prior-fitting for causal effect estimation below.

**Causal Structures**  Causality can be formalised via structural causal models (SCMs; Pearl, 2009; Peters et al., 2017). They consist of three main components: **(i)** a directed acyclic graph (DAG) $\mathcal{G} = (\mathcal{N}, \mathbf{A})$, where the set of nodes $\mathcal{N} = \{z_1, \ldots, z_K\}$ corresponds to random variables and the structure of the graph is encoded by an adjacency matrix $\mathbf{A} \in \{0, 1\}^{K \times K}$. An entry $A_{ij} = 1$ indicates the presence of a directed causal edge $z_i \rightarrow z_j$, while $A_{ij} = 0$ indicates the absence of a direct causal influence. The adjacency matrix thus determines the parent set of each variable as $\text{pa}(z_k) = \{z_j \mid A_{jk} = 1\}$. **(ii)** SCMs pose a collection of causal mechanisms $\{f_k\}_{k=1}^K$, where each function $f_k$ determines the value of $z_k$ as a function of its parents $\text{pa}(z_k)$ and an exogenous noise variable $\varepsilon_k$, i.e., $z_k := f_k(\text{pa}(z_k), \varepsilon_k)$, and **(iii)** a joint distribution $p(\varepsilon_1, \ldots, \varepsilon_K)$ over the exogenous noise variables, which we assume to be mutually independent. We denote an SCM as the tuple $\psi := (\mathcal{G}, \{f_k\}_{k=1}^K, p(\varepsilon_1, \ldots, \varepsilon_K))$ and assume no hidden variables, i.e., causal sufficiency, throughout.

SCMs can act as generative models via ancestral sampling; i.e., first drawing the exogenous variables, followed by computing the functional mechanisms of the SCM in a topological order of the nodes. This defines an observational distribution $p(\mathcal{D} \mid \psi)$, where $\mathcal{D} = \{(\boldsymbol{x}_{tr}^{(i)}, y_{tr}^{(i)})\}_{i=1}^{N_{tr}}$ comprises $N_{tr}$ i.i.d. samples obtained via ancestral sampling, where one (randomly determined) node $z_i$ represents the target $y_{tr}$ and the other nodes the feature-vector $\boldsymbol{x}_{tr}$. Later, we will use ancestral sampling to generate synthetic data.

**Interventions and Interventional Distributions**  SCMs define not only the observational distribution but also distributions under *interventions*, formalised by the do-operator $\text{do}(\cdot)$. An intervention $\text{do}(t)$ replaces the structural assignment of variable $t^2$ with a constant, thereby removing all incoming edges into $t$ in the underlying DAG. We denote an SCM that has been intervened upon with respect to variable $t$ by $\psi_{\text{do}(t)}$. This operation induces the conditional interventional distribution (CID; Robins 1986; Shpitser & Pearl 2006) $p(y \mid \boldsymbol{x}, \psi_{\text{do}(t)})$, which describes, for a fixed SCM $\psi$, the distribution of an outcome $y$ under an intervention on $t$, conditional on observed features $\boldsymbol{x}$.

Access to the SCM $\psi$ enables the computation of interventional queries with observational data (Pearl, 2009). In the absence of $\psi$, estimating the SCM from observational data is typically ill-posed: multiple SCMs can imply the same observational distribution, and point identifiability requires restrictive functional assumptions (Peters et al., 2017). These hard restrictions on functions can lead to poor performance if they do not hold. In contrast, the Bayesian approach allows for replacing the strict functional constraints with soft constraints through a prior $p(\psi)$, with the posterior $p(\psi|\mathcal{D})$ capturing the uncertainty due to the lack of identifiability. It has been shown that under the relatively weak assumption that the prior over functional mechanisms factorises[3], the posterior over causal structure cannot be made noninformative (Dhir et al., 2024). This allows the posterior to distinguish causal direction, albeit with some theoretical probability of error (Dhir et al., 2024), and can offer better performance compared with making restrictive functional assumptions for point identifiability (Dhir et al., 2025c;a). We incorporate factorised priors throughout our work.

---

[2]We make use of a common abuse of notation by using $t$ to denote both the variable representing $t$ and its value.

[3]This is an instantiation of the *independent causal mechanism* principle (Stegle et al., 2010; Janzing & Schölkopf, 2010; Guo et al., 2024).

The uncertainty in the posterior can be propagated to the interventional distribution by averaging the posterior over SCMs (Madigan & Raftery, 1994).

$$p(y|\text{do}(t), \boldsymbol{x}, \mathcal{D}) = \int p(y|\boldsymbol{x}, \psi_{\text{do}(t)}) p(\psi|\mathcal{D}) d\psi. \quad (1)$$

We refer to the posterior predictive in Equation (1) as the CID throughout the paper. Computing Equation (1) is difficult due to the intractable integral, especially for complex priors $p(\psi)$ and therefore requires approximations, for instance, via Neural Processes.

**Prior-Fitting for Causal Effect Estimation**  Neural processes are models that meta-learn to map from datasets to a complex, usually intractable, posterior predictive distribution of interest (Garnelo et al., 2018a;b; Nguyen & Grover, 2022). Here, the training datasets form the prior distribution and the approximation of the posterior predictive sidesteps any intermediate quantities. Prior-data fitted Networks (PFNs; Müller et al. 2021) are a class of neural processes that learn solely from synthetic datasets sampled from a carefully constructed prior. PFNs thus are well suited for approximating Equation (1): sampling interventional data from a distribution over SCMs is simple, and the approximation directly estimates Equation (1), without explicitly working with the potentially complex posterior over SCMs. Training data is sampled by **1)** sampling an SCM from a prior $\psi \sim p(\psi)$, followed by **2)** sampling an observational dataset $\mathcal{D} \sim p(\mathcal{D}|\psi)$ from the observational distribution via ancestral sampling, and **3)** sampling interventional testpoints from the SCM $(\boldsymbol{x}, y) \sim p(\boldsymbol{x}, y \mid \psi_{\text{do}(t)})$ by drawing a value for the intervention $t \sim p(t \mid \psi)$, followed by propagating it through the intervened upon SCM $\psi_{\text{do}(t)}$ together with newly sampled noise. This sampling procedure defines a joint distribution over observational-interventional data $p(\mathcal{D}, t, \boldsymbol{x}, y)$. CFMs can be trained in the PFN-paradigm by minimising the negative log-likelihood on data sampled from the prior. This corresponds to minimising the forward KL-divergence between the model's approximation $q_{\boldsymbol{\theta}}(y|\text{do}(t), \boldsymbol{x}, \mathcal{D})$ and the CID from Equation (1):

$$\underset{\theta}{\text{argmin}} \, \mathbb{E}_{p(\mathcal{D}, t, \boldsymbol{x}, y)} - \log q_{\boldsymbol{\theta}}(y|\text{do}(t), \boldsymbol{x}, \mathcal{D}) =$$

$$\underset{\theta}{\text{argmin}} \, \mathbb{E}_{p(\mathcal{D}, t, \boldsymbol{x})} \mathbb{KL} \left[ p(y|\text{do}(t), \boldsymbol{x}, \mathcal{D}) || q_{\boldsymbol{\theta}}(y|\text{do}(t), \boldsymbol{x}, \mathcal{D}) \right]$$

$$(2)$$

We refer to Robertson et al. (2025) and Dhir et al. (2025b) for a detailed explanation of PFNs and Neural Processes for causal effect estimation.

## 3. Related Work

To the best of our knowledge, all existing Causal Foundation Model (CFM) approaches operate in the PFN-paradigm, with a distinction into two types of methods: 1) models that are only trained on data from SCMs where the interventional query is identifiable from data—a separate model is required for each identifiable case, reflecting different causal assumptions, and 2) models that do not impose any assumptions and rely entirely on the posterior over SCMs.

Ma et al. (2025) propose using separate priors $p_1(\psi), p_2(\psi), \ldots, p_N(\psi)$ for different causal settings (back-door adjustment, front-door adjustment, and instrumental variables). A different model $q_{\theta_1}, q_{\theta_2}, \ldots, q_{\theta_N}$ is trained on data from each individual prior. This ensures by a restriction of the possible graph-types per prior, that a supposed "ground-truth" causal effect can be recovered in the limit of infinite data. Furthermore, CausalPFN (Balazadeh et al., 2025) is a PFN trained for estimating causal effects only in the back-door case (more specifically under strong ignorability), demonstrating excellent performance on benchmarks that satisfy that assumption. The above models presuppose access to causal information that renders the interventional distribution point-identifiable. In the absence of this, these models are misspecified and cannot be used for estimating interventional distributions. In contrast, methods like DoPFN (Robertson et al., 2025), MACE-TNP (Dhir et al., 2025b), and ACTIVA (Sauter et al., 2025) do not impose restrictions on the distribution over causal graphs and train on a single prior $p(\psi)$. These methods estimate the uncertainty due to non-identifiability, and also take advantage of the fact that the Bayesian posterior can be informative about the SCM in Markov-equivalent cases by, e.g., relying on the independence of mechanisms (Dhir et al., 2025c;b). However, estimates from these models can still include uncertainty from SCMs that can easily be rejected through domain knowledge. As exemplified in Figure 2, when domain knowledge is available, these models are overly conservative with their estimates, because they omit potentially critical causal information.

We argue that an all-in-one causal foundation model requires the ability to condition on both observational data **and** any available causal information when making predictions. A major challenge for such an approach is how to effectively condition on causal information—which is the question we tackle in this paper.

## 4. Methodology

To address the problem of conditioning on causal information in CFMs, we aim to answer two questions: **(a)** What type of causal information should one condition on in CFMs? **(b)** How to condition on causal information in CFMs? We begin by arguing in favour of partially known ancestral matrices as an answer to question **(a)**:

## 4.1. Ancestral Relationships

Beyond the direct causal connections specified via an adjacency matrix (introduced in Section 2), we can characterise the broader dependency structure of an SCM through its ancestor relationships. A variable $z_i$ is said to be an ancestor of $z_j$ if there exists a (possibly multi-edge) directed path $z_i \rightsquigarrow z_j$ in the DAG. The totality of those relationships can be represented by the *ancestor matrix* $\mathbf{T} \in \{0, 1\}^{K \times K}$, defined such that $T_{ij} = 1$ if and only if $z_i$ is an ancestor of $z_j$ (and 0 otherwise). Formally, $\mathbf{T}$ corresponds to the transitive closure of the adjacency matrix $\mathbf{A}$, i.e. $\mathbf{T} = \text{sgn}(\mathbf{A} + \mathbf{A}^2 + \mathbf{A}^3 + \ldots)$. Ancestor matrices generally contain less information than adjacency matrices since multiple adjacency matrices can yield the same ancestral matrix (Aho et al., 1972). However, this provides them with practical advantages over working with adjacency matrices:

While adjacency matrices require a practitioner to know for all pairs of variables $(z_i, z_j)$ if $z_i$ is a direct cause of $z_j$, ancestral matrices only require the assumption that $z_i$ is a direct *or indirect* cause of $z_j$. It is, for example, uncontroversial to state that smoking causes cancer; however, assuming that smoking is a *direct cause* is no longer correct when closely inspecting the biological mechanisms (The Surgeon General, 2010). Furthermore, one can argue that ancestor matrices asymptotically contain the same causal information as adjacency matrices under standard assumptions. This is because from an infinite observational dataset, one can determine the skeleton of the ground-truth causal DAG under standard conditions (Theorem 3.4 in Spirtes et al. (2000)), while checking for all pairs $(i, j)$ if $z_i$ is ancestor of $z_j$ yields the remaining edge orientations.

## 4.2. Partial Ancestral Information

While knowing ancestral instead of direct causal relationships has the meaningful practical benefits discussed above, some (or even all) of the ancestral relationships within a set of variables may be unclear. We thus propose to utilise the partially known ancestor matrix (PAM), denoted by $\tilde{\mathbf{T}} \in \{-1, 0, 1\}^{K \times K}$. Specifically, we set $\tilde{T}_{ij} = 1$ if it is known that $z_i$ is a cause of $z_j$ and $\tilde{T}_{ij} = -1$ if it is known that $z_i$ is not a cause of $z_j$. Importantly, we set $\tilde{T}_{ij} = 0$ if the causal relationship between $z_i$ and $z_j$ is unknown.

Having partial causal knowledge that cannot be specified in a fully known causal graph is common in real-world datasets. As an example, we discuss how this applies to the IHDP dataset in Appendix C.

Furthermore, the temporal order always provides a natural constraint on ancestral causal structure, since causes must precede effects. If variable $z_i$ is measured before $z_j$ in time, we immediately know that $\tilde{T}_{ji} = -1$, ruling out backward causation. In practice, this type of temporal information,

at least for a subset of all available variables, is arguably a very common type of causal information.

Considering classical causal assumptions, it is straightforward to specify back-door and front-door constellations via ancestral relationships and thus PAMs. As an additional example, we discuss how *unconfoundedness*, an assumption commonly made in the potential outcomes framework (PO; Imbens & Rubin 2015), can be specified using PAMs in Proposition A.1. Later, we show that encoding unconfoundedness via a PAM provides meaningful benefits to the predictions of a CFM (Section 5.5).

**Conditioning on Graphs and Consistency** Ma et al. (2025) argue in favour of priors $p(\psi)$ that ensure consistency of the induced posterior $p(y|\text{do}(t), \boldsymbol{x}, \mathcal{D})$[4] by restricting the prior over graphs to specific identification setups, followed by training separate models for each case. However, graph-conditioning allows training a single model on a generic prior, while consistency is (trivially) implied whenever the graphical information suffices for identifiability. If the causal effect cannot be identified via partial graph knowledge, the posterior quantifies the uncertainty due to unidentifiability. We provide a mathematical discussion of consistency using (partial) graph information in our Bayesian setup in Appendix B.

## 4.3. Architectures for Conditioning on Graphs in Causal Foundation Models

Now, we discuss architectural modifications for conditioning on graphical causal information in the form of partially known ancestral matrices, introduced in Section 4.2.

**Graph-Conditioned Architecture** Following Dhir et al. (2025b) and Robertson et al. (2025), we employ a transformer-based architecture (Vaswani et al., 2023). We begin by embedding the input data such that the model (i) can capture intra-sample correlations, (ii) preserves the specific value of each variable, and (iii) can distinguish between different variable roles (e.g. feature vs. treatment) and data types (observational vs. interventional). The exact embedding procedure is detailed in Appendix D.

We then apply multiple transformer blocks with a factorised attention mechanism that alternates between two operations at each layer: (i) sample-wise attention, which exchanges information across different samples, and (ii) feature-wise attention, which models relationships between variables within a sample. In the feature-wise stage, we condition the model on the partial ancestral matrix $\tilde{\mathbf{T}}$, and propose two complementary mechanisms for injecting this structural

---

[4]Note that properties of this posterior do not necessarily translate to the approximation $q_\theta(y|\text{do}(t), \boldsymbol{x}, \mathcal{D}) \approx p(y|\text{do}(t), \boldsymbol{x}, \mathcal{D})$ learned by the neural network.

information: *structural attention biasing* and *structural feature modulation*.

**Structural Attention Biasing**   The feature-wise attention blocks in our model compute attention scores between all $K$ variables (features) in a dataset, in order to exchange information about the representation of each feature. The attention scores can be summarised in a matrix $\mathbf{A} \in \mathbb{R}^{K \times K}$, where entry $A_{i,j}$ quantifies how much attention feature $i$ (query) pays to feature $j$ (key):

$$A_{i,j} = \frac{\exp\left(\frac{\mathbf{q}_i^\top \mathbf{k}_j}{\sqrt{d_k}}\right)}{\sum_{j'=1}^{K} \exp\left(\frac{\mathbf{q}_i^\top \mathbf{k}_{j'}}{\sqrt{d_k}}\right)}, \tag{3}$$

where $\mathbf{q}_i$ denotes the query vector corresponding to feature $i$, $\mathbf{k}_j$ the key vector of feature $j$, and $d_k$ the dimensionality of the key vectors.

To inject causal knowledge, we bias the attention scores in the direction of the ancestral causal relationships between the variables, exploring two variants:

**1.) Soft Attention Bias:** We introduce learnable scalar biases that encourage attention towards known ancestors and discourage attention towards known non-ancestors. The attention logits are modified by adding $B_{ij}$ to $A_{ij}$ to obtain $\widetilde{A}_{i,j} := A_{ij} + B_{ij}$, where

$$B_{ij} = \begin{cases} +\beta_{\text{anc}} & \text{if } z_j \text{ ancestor of } z_i & (\tilde{T}_{ji} = 1) \\ -\beta_{\text{non-anc}} & \text{if } z_j \text{ non-ancestor of } z_i & (\tilde{T}_{ji} = -1) \\ 0 & \text{if unknown} & (\tilde{T}_{ji} = 0). \end{cases}$$

Here, $\beta_{\text{anc}}$ and $\beta_{\text{non-anc}}$ are learnable scalar bias terms that increase or decrease attention between features depending on their ancestral relationship, and we use separate learnable parameters per head per layer. (E.g., for a model with six layers and four attention heads, we would use 24 additional parameters). The matrix $\widetilde{\mathbf{A}}$ is then used to compute the result of the attention computation: $\text{Attention}(\mathbf{Q}, \mathbf{K}, \mathbf{V}) = \widetilde{\mathbf{A}}\mathbf{V}$, where $\mathbf{V} \in \mathbb{R}^{K \times d_v}$ denotes the matrix of value vectors. This soft formulation allows the model to prioritise causal parents while remaining robust to incomplete graph information. Note that we add the bias $\beta_{\text{anc}}$ to the attention score between the $i$-th and $j$-th variable if $z_j$ is ancestor of $z_i$ (and not if $z_i$ is ancestor of $z_j$), as this encourages the flow of information in the causal direction.

**2.) Hard Masking:** Alternatively, we enforce sparsity by setting $B_{ij} = -\infty$ if $\tilde{T}_{ji} = -1$, and $B_{ji} = 0$ otherwise. This strictly prohibits information flow from variables known not to be causes while treating ancestor and unknown node relationships equally.

**Structural Modulation (GCN)**   Complementary to local attention biasing, we condition the model on a global representation of the ancestry matrix $\tilde{\mathbf{T}}$ using a Graph Convolutional Network (GCN; Duvenaud et al. 2015) adapter. While the attention biases described above act directly on pairwise interactions, the GCN provides each variable with a structure-aware representation that captures its broader role in the causal graph. Concretely, we initialise each node with a learnable role embedding indicating whether the corresponding variable is a feature, treatment, or outcome, and propagate these embeddings over the graph induced by $\tilde{\mathbf{T}}$ to obtain graph-conditioned node representations $\mathbf{Z}_{\text{graph}}$. These are injected into each transformer block via Adaptive Layer Normalisation (AdaLN; Peebles & Xie 2023):

$$\widetilde{\mathbf{X}} = \boldsymbol{\gamma}(\mathbf{Z}_{\text{graph}}) \odot \text{LN}(\mathbf{X}) + \boldsymbol{\delta}(\mathbf{Z}_{\text{graph}}),$$

where $\mathbf{X}$ denotes the transformer hidden representations. This allows the hidden representation of each variable to be modulated according to its structural position in the causal graph, enabling the model to incorporate global graph information throughout the network.

**Sample-wise Attention and Decoding**   After graph-conditioned feature processing, the model performs sample-wise attention to aggregate information across the dataset. We implement this using Multi-Head Self-Attention over the observational data, followed by Multi-Head Cross-Attention from the query to the observational set. This process is repeated over $L$ blocks. Finally, we extract the outcome representation for the query sample and project it through a linear head to parameterise a discretised bar distribution (Hollmann et al., 2022) over the outcomes. Complete implementation details are provided in Appendix D.

### 4.4. A Tabular Foundation Model Prior with a Causal Implementation

To provide a challenging and realistic test-bed for training and evaluating the effect of graph-conditioning, we turn to priors, i.e., simulators of synthetic tabular data, from state-of-the-art Tabular Foundation Models (TFMs; Hollmann et al. 2022; Qu et al. 2025). Since no existing TFM prior provides a natively causal implementation (also see the discussion in Appendix E.1), we built upon the prior in (Robertson et al., 2025) and implemented advancements from the state-of-the-art open-source prior in TabICL in terms of functional mechanisms (Qu et al., 2025). We show that training a TFM on this prior yields competitive predictive performance compared to, e.g., a Random Forest with default hyperparameters (Breiman, 2001), on small-scale tabular regression benchmark datasets (details in Appendix E.2). In the subsequent experiments, to evaluate the effect of graph-conditioning, we use exactly this prior ("TFM-Prior"), as well as a linear-Gaussian version with exclusively linear mechanisms and Gaussian noise ("LinGaus-Prior"). Our prior implies the independence of noise variables, indepen-

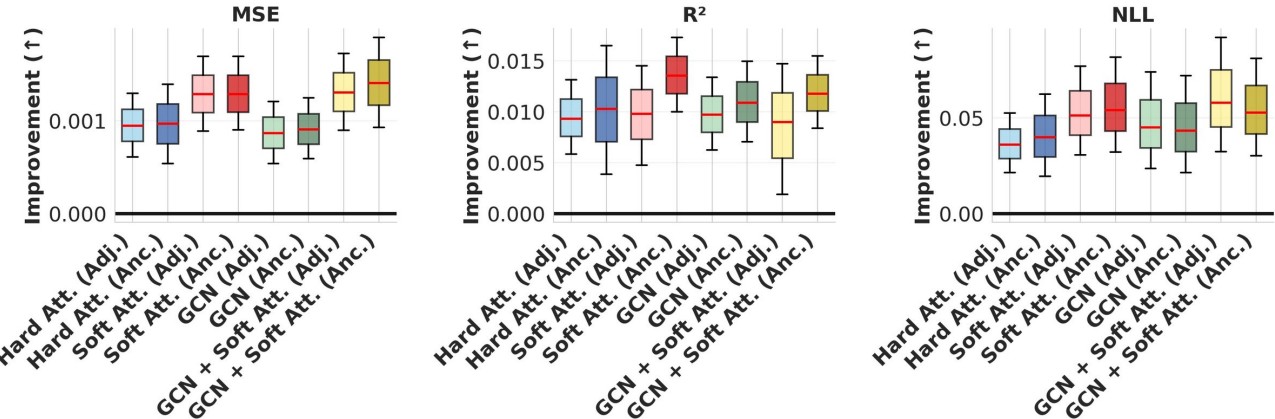

*Figure 3.* Performance comparison of different graph-conditioning methods on linear-Gaussian data. Results are shown as the improvement in negative log-likelihood (NLL), mean-squared-error (MSE) and coefficient of determination ($R^2$) on held-out test data relative to a non-graph-conditioned baseline (represented by the value 0 on the $y$-axis). The error-bars represent 95-percent bootstrap confidence intervals for the median. We compare soft and hard attention biases as well as a GCN-based conditioning approach, using either ground-truth adjacency matrices (Adj.) or ancestor matrices (Anc.) as graph input. All methods yield significant benefits with the Soft Attention and GCN + Soft Attention methods performing the best.

dence of causal mechanisms, as well as causal sufficiency, i.e., no hidden variables. A somewhat surprising finding is that the TFM-Prior yields relatively small absolute differences between causal and observational posteriors when performing hard interventions (see Appendix E.3).

# 5. Experiments

We empirically study the effectiveness of graph-conditioning in Causal Foundation Models (CFMs), progressing from controlled synthetic settings to complex and semi-synthetic benchmarks. In all our experiments, we assume no hidden variables, i.e., causal sufficiency.

## 5.1. Which Graph-conditioning Method Works Best?

Our primary objective is to enable Causal Foundation Models (CFMs) to condition effectively on ancestral knowledge. As outlined in Section 4.3, this can be achieved through several distinct mechanisms, and our first research question seeks to identify the most effective approach. To isolate the contribution of the graph information, we first benchmark these methods on synthetic case studies where the graph information is critical because the underlying causal graph is not identifiable from the observational data alone. To achieve this, we train and evaluate a model on the LinGaus Prior (see Appendix F.1). Then we evaluate on a test-set sampled from the same distribution.

Figure 3 presents the improvement in Negative Log-Likelihood (NLL), Mean-Squared Error (MSE) and the $R^2$ score relative to a baseline that is trained without graph-conditioning but otherwise identical. We also evaluate two types of conditioning information: the ground-truth adja-

cency matrix (Adj.) and the ancestor matrix (Anc.). (see Section 4.1 for more details). As shown in Figure 3, all graph-conditioned variants outperform the baseline, confirming that the model effectively leverages structural information. Among the investigated architectures, Soft Attention and the combination of Soft Attention with a GCN consistently outperform both Hard Attention and the GCN-only variants. When comparing conditioning signals (Adj. vs. Anc.), we generally only observe a small divergence in performance, empirically confirming that CFMs can be effectively conditioned on ancestor information. Interestingly, using the ancestor matrix instead of the adjacency matrix yields small benefits in terms of MSE and $R^2$, especially for the soft-attention-based methods. We hypothesise that restricting attention solely to direct parents (as implied by the adjacency matrix) may be overly limiting for bias-based mechanisms, which benefit from the broader context provided by the ancestor matrix. Detailed results can be found in Appendix H.1. While the observed effect of graph-conditioning compared to the baseline is always significant, it is small in absolute terms across all metrics. However, this is in line with overall small causal effects under our TFM prior (Appendix E.3).

## 5.2. Is Graph-Conditioning still Effective with Partial, rather than Full Ancestry Information?

In order to support no, full and partial conditioning, we need to amortise during training over scenarios with varying amounts of available ancestor information. This raises the question of how much performance the model loses due to the resulting challenges this entails. To obtain an answer, we compare a model capable of handling any amount of

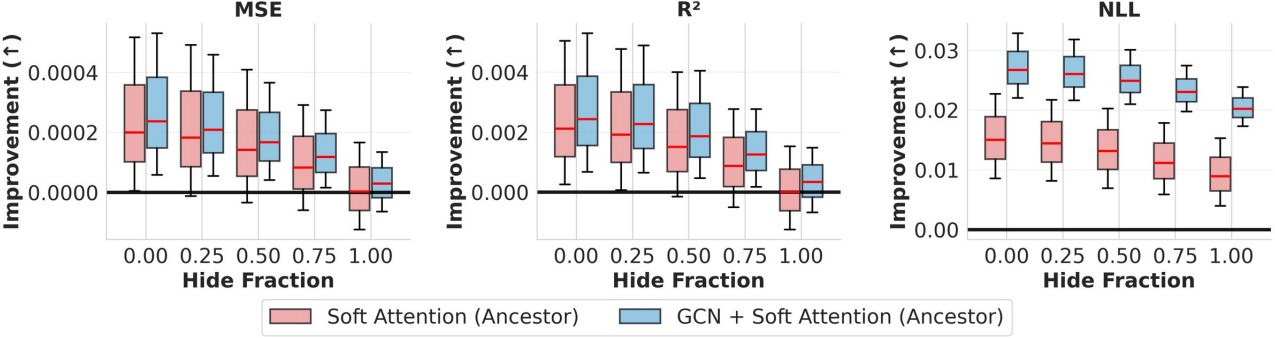

*Figure 4.* Performance on complex synthetic data for different fractions of hidden entries in the ancestor matrix for the Soft Attention (Ancestor) and GCN + Soft Attention (Ancestor) graph-conditioning approaches. Results are measured as the absolute difference to a baseline that does not support graph conditioning but is otherwise trained identically. The baseline performance is represented by the value 0 on the $y$-axis. Error-bars represent 95-percent bootstrap confidence intervals for the mean. Providing full graph information significantly improves performance while GCN + Soft Attention performs better than just using Soft Attention.

missing information (amortised) with specialised models: one specialised in never using ancestral information; one specialised in always using full ancestral information. As observed in Figure 13 in Appendix H.2, the amortised procedure during training is efficient, with the performance of the amortised model being comparable to that of the specialised models. This shows that our training procedure yields a model that is flexible concerning the amount of information it receives without sacrificing performance—a single model can effectively handle varying amounts of available information.

### 5.3. Does Graph-conditioning Help with Complex Causal Structures?

Besides the simple linear-Gaussian mechanisms that we considered in the previous sections, we now turn to the more complex and realistic case of the TFM prior. We evaluate the two best graph-conditioning mechanisms from Section 5.1, namely Soft Attention and Soft Attention combined with a GCN on 30,000 datasets from the complex prior. We vary the fraction of entries in the ancestor matrix that are hidden, i.e., replaced by 0, and measure the absolute improvement compared to a model that is not trained with ancestor-conditioning, but otherwise identical.

First, Figure 4 shows that providing full graph-information improves the error of the model in all metrics and for both graph-conditioning methods. When we increase the fraction of hidden entries in the ancestor matrix towards one, the models' performance degrades and is, for MSE and $R^2$, no longer significantly different from the non-graph-conditioned baseline. The NLL metric indicates a consistent benefit of training models with ancestral information on the complex mechanisms, even when no ancestral information is provided. We hypothesise that this is due to improved training dynamics when the causal structure of the data is

respected during training. In terms of the NLL, one can observe that the combined GCN + Soft Attention approach performs noticeably better than just using Soft Attention, while in terms of the less sensitive MSE and $R^2$ metrics, which assess only the mean of the PPD, there is a systematic yet not significant difference. Overall, the improvement relative to the baseline remains small in terms of absolute values, which is again in line with small differences between the interventional and observational posteriors (Appendix E.3).

### 5.4. Out-of-Distribution Performance

A natural concern for graph-conditioned models is their robustness to errors in the provided ancestral information. As shown in Appendix K, when edges are misspecified (i.e., the stated causal direction is incorrect), performance degrades quickly as the fraction of incorrectly specified edges grows—an expected finding that is equally true for traditional causal methods. However, a key practical advantage of our approach is that practitioners need *not* commit to a potentially wrong causal direction: uncertain relationships can instead be marked as "unknown" by setting the corresponding entry to 0. We find that this strategy is substantially less harmful than misspecification, with performance degrading only gradually as more edges are hidden. This highlights the practical benefit of supporting partial ancestral matrices: when causal domain knowledge is uncertain, leaving the relevant edges unspecified is far better than guessing incorrectly.

### 5.5. Does Partially Known Ancestral Information Help on Semi-Synthetic Data?

Finally, we also investigate whether providing our model with graph information offers benefits on the commonly used semi-synthetic RealCause Benchmark (Neal et al., 2020). The data for this benchmark is simulated to satisfy the unconfoundedness assumption, which does not spec-

*Table 1.* Providing graph information helps for semi-synthetic data: We evaluate the effect of providing partially known ancestral information encoding unconfoundedness on semi-synthetic RealCause datasets. We report the performance for a **predictive** model trained on our complex prior, a model with **partially known ancestral information**, and a model with **no ancestral information**. As metrics, we use root Precision in Estimation of Heterogeneous Effect ($\sqrt{\text{PEHE}}$) and relative error when predicting the ATE $\epsilon_{\text{ATE}}$ (see Appendix F.2.1).

| Method | IHDP | | ACIC | | CPS | | PSID (unbalanced) | |
|---|---|---|---|---|---|---|---|---|
| | $\sqrt{\text{PEHE}}$ ($\downarrow$) | $\epsilon_{\text{ATE}}$ ($\downarrow$) | $\sqrt{\text{PEHE}}$ ($\downarrow$) | $\epsilon_{\text{ATE}}$ ($\downarrow$) | $\sqrt{\text{PEHE}}$ ($\downarrow$) | $\epsilon_{\text{ATE}}$ ($\downarrow$) | $\sqrt{\text{PEHE}}$ ($\downarrow$) | $\epsilon_{\text{ATE}}$ ($\downarrow$) |
| Predictive | 6.79±0.81 | 0.81±0.11 | 3.14±0.47 | 0.38±0.06 | 11393±31 | 0.78±0.00 | **11820**±41 | 1.03±0.01 |
| No Ancestral Info. | 6.28±0.79 | 0.67±0.05 | 3.47±0.47 | 0.46±0.09 | 12800±55 | 0.99±0.01 | 13096±26 | **0.98**±**0.00** |
| Ancestral Info. | **5.49**±**0.78** | **0.49**±**0.08** | **2.79**±**0.45** | **0.17**±**0.08** | **11213**±**60** | **0.70**±**0.02** | 12975±24 | 1.09±0.01 |

ify the causal relationship between the covariates. This makes it impossible to directly use complete adjacency matrices or full ancestral matrices to exactly represent this assumption; however, it can be specified in terms of partially known ancestral matrices (Proposition A.1). To evaluate the RealCause benchmark, we train a model on the complex-mechanism prior from Section 5.3 and assess the effect of providing ancestral information. Fixing the prior while using a model that (a) is trained in the predictive setup, (b) uses the ancestral information, and (c) does not use the ancestral information eliminates the confounding effect of different priors, thus allowing for a precise assessment of the effect of conditioning on ancestor information. We compare the performance of the model without any graph information ("No Ancestral Info.") to the model with the correct graph information ("Ancestral Info."). Furthermore, we compare against a predictive model ("Predictive") trained on the complex-mechanisms prior, as under unconfoundedness, predicting the causal effect effectively becomes a predictive task (Künzel et al., 2019).

Table 1 shows that providing the model trained on the generic complex prior with the partially known ancestral matrix systematically improves performance compared to not conditioning on ancestral information for the IHDP, ACIC and CPS datasets. For PSID, all methods achieve suboptimal performance, which we find is due to the highly imbalanced distribution of treated versus untreated individuals. When performing rebalancing, all methods substantially improve while conditioning on the ancestral information yields substantial benefits (see Appendix F.3).

### 5.6. Comparison to Other Causal Foundation Models

Unlike our approach, CausalPFN (Balazadeh et al., 2025) and CausalFM (Ma et al., 2025) are trained exclusively for specific identification settings. We compare our models against these baselines on the TFM prior, stratified by whether the backdoor criterion holds (Appendix J and Figure 18), which is assumed by both CausalPFN and (this version of) CausalFM. Our graph-conditioned models perform only slightly better than CausalPFN when the backdoor criterion is satisfied; when it is violated—a setting where CausalPFN and CausalFM rely on a misspecified assumption—the advantage of our models increases substantially. On the RealCause benchmark (Table 3), CausalPFN performs best overall, which can be attributed to its training process being designed precisely for this case (Neal et al., 2020; Balazadeh et al., 2025); once graph conditioning is introduced, our model outperforms CausalFM and Do-PFN (Robertson et al., 2025) across almost all datasets.

## 6. Discussion

In this work, we study how to condition CFMs on structured causal knowledge. We argue that partially known ancestral matrices provide a realistic and expressive way to encode domain knowledge, capturing the common case where some causal relationships can be known, some ruled out, and some uncertain. Architecturally, we identify that the most efficient method to inject this information is by combining attention biasing with a graph convolutional encoder. Using this strategy, we enable the construction of a single CFM that accounts for uncertainty when specific causal domain knowledge is lacking, while effectively leveraging varying amounts of available causal information, which we demonstrate on complex, non-identifiable synthetic data as well as semi-synthetic benchmarks.

**Limitations** This work focuses on graphical assumptions regarding fully observed variables encoded through partially known ancestry. We do not consider the case of hidden variables. While partial graph information is a widely applicable form of causal knowledge, CFMs could also benefit from flexibly conditioning on aspects of functional mechanisms or hidden variables. Moreover, while we evaluate on a complex causal prior, validated in the predictive setting, the development of practically useful "all-in-one" CFMs would require comprehensive real-world causal benchmarks that currently do not exist. Despite these current constraints, our method successfully demonstrates that conditioning on causal information is both feasible and effective. This establishes a promising foundation for the next generation of CFMs, bringing us closer to robust, all-in-one causal effect estimation models.

## Acknowledgements

AR is supported by the Leverhulme Trust via the Leverhulme Centre for the Future of Intelligence. AW acknowledges support from a Turing AI fellowship under grant EP/V025279/1, The Alan Turing Institute, and the Leverhulme Trust via CFI.

## Impact Statement

This paper presents work whose goal is to advance the field of Machine Learning. There are many potential societal consequences of our work, none of which we feel must be specifically highlighted here.

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

# A. Detailed Propositions

**Proposition A.1** (Unconfoundedness from partial graph knowledge). *Assume a causally sufficient SCM on the observed variables $\mathcal{N} = \{t, y, x_1, \ldots, x_K\}$ with potential outcomes $y_0, y_1$, a binary treatment $t \in \{0, 1\}$, and*

$$y = ty_1 + (1 - t)y_0.$$

*Let $\tilde{\mathbf{T}}$ be a PAM such that:*

- *$t \rightsquigarrow y$ is specified in the PAM,*

- *for all $i$, $x_i \rightsquigarrow t$ and $x_i \rightsquigarrow y$ are specified in the PAM,*

- *no variable outside $\boldsymbol{x} = \{x_i\}_{i=1}^{K}$ is allowed to be an ancestor of both $t$ and $y$ (i.e., the PAM excludes any common ancestor of $t$ and $y$ outside $x$).*

*Then the unconfoundedness condition holds:*

$$(y_0, y_1) \perp\!\!\!\perp t \mid x.$$

*Proof sketch.* The PAM constraints enforce that all common causes of $t$ and $y$ are contained in $\boldsymbol{x}$. Since the model is causally sufficient, there are no unobserved confounders. Conditioning on $\boldsymbol{x}$ therefore removes all shared sources of variation between the treatment assignment mechanism and the outcome mechanism. Intervening on $t$ changes only the structural equation of $t$ and does not introduce new dependence between $t$ and $(y_0, y_1)$ once $\boldsymbol{x}$ is fixed. Hence, conditional on $\boldsymbol{x}$, treatment assignment is independent of the potential outcomes, which is exactly the unconfoundedness assumption $(y_0, y_1) \perp\!\!\!\perp t \mid \boldsymbol{x}$. $\square$

*Remark* A.2. Not assuming a path between $x_i$ and $t$, or between $x_i$ and the outcome $y$, can equally be expressed in the PAM by leaving those entries unspecified (set to 0), and the unconfoundedness conclusion still follows. The only critical condition is that no variable outside $\boldsymbol{x}$ is a common ancestor of both $t$ and $y$.

# B. A Discussion on Consistency through Graphical Information

Suppose a ground-truth DAG $\mathcal{G}^*$ with ground-truth mechanisms and distribution parameters $\theta^*$ fully specifying an SCM $\psi^*$. Assume we have an observational dataset $\mathcal{D}_n$ of size $n$ sampled from $P(\mathcal{D}_n|\psi^*)$. Also, assume that we have correct, but potentially partial graph information $\mathcal{G}_0$ about $\mathcal{G}^*$, which, together with the parameter vector $\theta^*$, uniquely identifies the CID $P(y|\mathrm{do}(t), \boldsymbol{x}, \theta^*, \mathcal{G}_0)$, such that $P(y|\mathrm{do}(t), \boldsymbol{x}, \theta^*, \mathcal{G}_0) \neq P(y|\mathrm{do}(t), \boldsymbol{x}, \theta', \mathcal{G}')$ for any tuple $(\theta', \mathcal{G}') \neq (\theta^*, \mathcal{G}_0)$.

The question we are interested in is whether, for infinitely many observational data points $n \rightarrow \infty$, the CID $P(Y|\mathrm{do}(t), \boldsymbol{x}, \mathcal{D}_n, \mathcal{G}_0)$ will converge to identifiable CID $P(Y|\mathrm{do}(t), \boldsymbol{x}, \theta^*, \mathcal{G}_0)$.

To address this question, assume a parametrisation such that $\theta \mapsto P(\mathcal{D}_n|\theta, \mathcal{G}_0)$ is injective, which might require working in function-space. Consistency arguments from Bayesian nonparametrics yield (under their respective conditions):

$$P(\theta|\mathcal{G}_0, \mathcal{D}_n) \rightarrow \delta(\theta^*).$$

One can use Doob's theorem for this type of consistency; which requires $\theta$ to be finite-dimensional, and ensures consistency almost surely in terms of $P(\theta)$, but otherwise only needs measurability of the map $\theta \mapsto P(\mathcal{D}_n|\theta, \mathcal{G}_0)$. See Miller (2018) (Theorem 2.4). One can also make use of more powerful results such as Theorem 3 by De Blasi & Walker (2013), or Corollary 1 by Walker (2004), also used by Nagler (2023) in the context of PFNs, which ensures weak convergence almost-surely with respect to the data-generating distribution under the ground-truth SCM.

Now, under regularity conditions on the functional $P(\theta|\mathcal{G}_0, \mathcal{D}_n) \mapsto \int P(y|\mathrm{do}(t), \boldsymbol{x}, \theta, \mathcal{G}_0)P(\theta|\mathcal{G}_0, \mathcal{D}_n)d\theta$, this implies:

$$P(Y|\mathrm{do}(t), \boldsymbol{x}, \mathcal{D}_n, \mathcal{G}_0) \rightarrow P(Y|\mathrm{do}(t), \boldsymbol{x}, \theta^*, \mathcal{G}_0),$$

which is exactly consistency to the identifiable CID. If the partial graphical information $\mathcal{G}_0$, together with $\theta^*$ fully identifies the SCM $\psi^*$, one has $P(Y|\mathrm{do}(t), \boldsymbol{x}, \theta^*, \mathcal{G}_0) = P(Y|\mathrm{do}(t), \boldsymbol{x}, \psi^*)$, i.e., consistency in terms of the ground-truth SCM $\psi^*$.

One such regularity condition is continuity of $P(\theta|\mathcal{G}_0, \mathcal{D}_n) \mapsto \int P(y|\text{do}(t), \boldsymbol{x}, \theta, \mathcal{G}_0) P(\theta|\mathcal{G}_0, \mathcal{D}_n) d\theta$, which allows to use the Continuous Mapping Theorem (Theorem 2.3 in Van der Vaart (2000)). Alternatively, it suffices that $P(y|\text{do}(t), \boldsymbol{x}, \theta, \mathcal{G}_0) P(\theta|\mathcal{G}_0, \mathcal{D}_n)$ is a bounded and continuous function of $\theta$ by the Portmanteau Theorem (Lemma 2.2 in Van der Vaart (2000)).

## C. Example for Partial Ancestral Information

One of the most frequently used datasets to benchmark causal effect estimation is the Infant Health and Development Program (IHDP) dataset (Brooks-Gunn et al., 1992). Here, a partially known causal structure arises naturally from temporal, biological, and logical considerations. Maternal characteristics such as age, race, and endocrine or nervous system disorders clearly precede and can plausibly influence prenatal behaviors like cigarette consumption, implying $\tilde{T}_{\text{maternal\_age,cigarettes}} = 1$. Likewise, these maternal and prenatal factors are known to affect birth-related outcomes such as the child's bilirubin level ($\tilde{T}_{\text{cigarettes,bilirubin}} = 1$), whereas reverse effects such as $\tilde{T}_{\text{bilirubin,cigarettes}} = -1$ can be ruled out. Social and demographic variables such as maternal race or birthplace are fixed attributes and thus cannot be caused by medical conditions or behaviours ($\tilde{T}_{\text{race,medical}} = 1$, $\tilde{T}_{\text{medical,race}} = -1$). In contrast, many dependencies among medical covariates, such as between endocrine and nervous system conditions, remain uncertain.

## D. Details on the Architecture

Our architecture shares a lot of similarities with the architectures proposed in Robertson et al. (2025); Dhir et al. (2025b), while also allowing for conditioning on partial ancestral information.

We assume we observe $N_{tr}$ samples from the observational data $\mathcal{D} = \{\boldsymbol{x}_{tr}^{(i)}, t^{(i)}, y_{tr}^{(i)}\}_{i=1}^{N_{tr}}$ and we aim to predict the distribution of the interventional outcome $y$ given a set of interventional features $\boldsymbol{x}$ and a treatment value $t$: $p(y \mid \text{do}(t), \boldsymbol{x}, \mathcal{D})$. We parameterise the latter with a bar distribution following Robertson et al. (2025); Hollmann et al. (2022).

**Data Pre-Processing** The input to the model is comprised of 1) the observational data $\mathcal{D} = \{\boldsymbol{x}_{tr}^{(i)}, t^{(i)}, y_{tr}^{(i)}\}_{i=1}^{N_{tr}} \in \mathbb{R}^{N_{tr} \times (D+2)}$, where $D$ represents the maximum number of features (excluding the treatment and outcome) used during training, and 2) the interventional query ($\boldsymbol{x} \in \mathbb{R}^D$, $t \in \mathbb{R}$). If the dataset of interest contains fewer than $D$ features we pad the input with zeros.

**Embeddings** Before applying any attention mechanism, we aim to embed the data such that the model 1) is able to capture intra-sample correlations amongst features, 2) can preserve the specific value of each variable, and 3) is able of distinguishing between the different types of variables (i.e., feature vs. treatment) and data sources (observational vs. interventional). Let $\mathbf{X} \in \mathbb{R}^{S \times D}$ and $\mathbf{t} \in \mathbb{R}^{S \times 1}$ denote a batch of $S$ samples (where $S = N_{tr} + 1$) comprising both observational data and the interventional query. We first concatenate features and treatments to form the input tensor $\mathbf{V} = [\mathbf{X}; \mathbf{t}] \in \mathbb{R}^{S \times (D+1)}$. This input is processed to produce embeddings in $\mathbb{R}^{S \times (D+1) \times d_{\text{model}}}$, where $d_{\text{model}}$ is the hidden dimension of the model:

1. **Row Embeddings:** To capture intra-sample correlations, we process the entire feature vector of a sample simultaneously. The concatenated input $\mathbf{V}$ is projected via a Multi-Layer Perceptron ($\text{MLP}_{\text{row}} : \mathbb{R} \to \mathbb{R}^{d_{\text{model}}}$) to generate a global sample embedding. This embedding is then broadcast across the feature dimension to yield $\mathbf{E}_{\text{row}} \in \mathbb{R}^{S \times (D+1) \times d_{\text{model}}}$.

2. **Cell Embeddings:** To preserve the precise value of individual variables, we apply a separate MLP ($\text{MLP}_{\text{cell}} : \mathbb{R}^1 \to \mathbb{R}^{d_{\text{model}}}$) element-wise to each scalar entry in $\mathbf{V}$. To distinguish between variable types (feature vs. treatment) and data sources (observational vs. interventional), we add learnable role embeddings $\mathbf{e}_{\text{role}} \in \mathbb{R}^{d_{\text{model}}}$ to the output:

$$\mathbf{E}_{\text{cell}}^{(i,j)} = \text{MLP}_{\text{cell}}(\mathbf{V}_{ij}) + \mathbf{e}_{\text{type}}[j] + \mathbf{e}_{\text{source}}[i] \tag{4}$$

where $\mathbf{e}_{\text{type}}$ distinguishes between features ($1 \dots D$) and the treatment column ($D+1$), and $\mathbf{e}_{\text{source}}$ distinguishes between observational samples ($1 \dots N_{tr}$) and the query sample.

The final embedding $\mathbf{H}^{(0)} \in \mathbb{R}^{S \times (D+1) \times d_{\text{model}}}$ is a learnable weighted sum of these two representations:

$$\mathbf{H}^{(0)} = \lambda_{\text{row}} \cdot \mathbf{E}_{\text{row}} + \lambda_{\text{cell}} \cdot \mathbf{E}_{\text{cell}} \tag{5}$$

where $\lambda_{\text{row}}$ and $\lambda_{\text{cell}}$ are learnable scalar parameters initialized to control the relative importance of global context versus local values. Finally, we append a learnable token representing the target variable $y$, resulting in a final input sequence shape of $S \times (D + 2) \times d_{\text{model}}$.

**Feature Positional Encodings**  Inspired by Qu et al. (2025), we additionally inject feature positional encodings. Although the architecture already breaks the symmetry through the use of row embeddings, we further assure that we avoid representational collapse through sinusoidal positional encodings. For the sequence of $D + 2$ tokens (representing the $D$ features, the treatment indicator, and the target variable), we compute the encoding $\mathbf{P} \in \mathbb{R}^{(D+2) \times d_{\text{model}}}$ as:

$$\mathbf{P}(pos, 2i) = \sin\left(\frac{pos}{10000^{2i/d\text{model}}}\right), \quad \mathbf{P}(pos, 2i+1) = \cos\left(\frac{pos}{10000^{2i/d\text{model}}}\right) \tag{6}$$

These fixed encodings are broadcast across the batch and sample dimensions and added element-wise to the variable embeddings $\mathbf{H}^{(0)}$.

**Attention Sinks**  Following Grinsztajn et al. (2025), we prepend a set of $N_{\text{sink}}$ learnable dummy samples to the input sequence in order to improve optimisation stability and prevent the attention heads from collapsing when no relevant context is found. We instantiate these sinks as learnable parameters $\mathbf{S}_{\text{feat}} \in \mathbb{R}^{N_{\text{sink}} \times (D+1) \times d_{\text{model}}}$ (corresponding to features and treatment) and $\mathbf{S}_{\text{target}} \in \mathbb{R}^{N_{\text{sink}} \times 1 \times d_{\text{model}}}$ (corresponding to the target variable). These are concatenated to form complete dummy samples $\mathbf{S} = [\mathbf{S}_{\text{feat}}; \mathbf{S}_{\text{target}}]$ and prepended to the sample axis of the input batch.

After performing the above-mentioned modifications, the input to the transformer backbone $\tilde{\mathbf{H}}^{(0)}$ has dimensions $(N_{tr} + 1 + N_{\text{sink}}) \times (D + 2) \times d_{\text{model}}$.

**Graph Conditioning Mechanism**  The main contribution of our work is enabling a causal foundation model to condition on partial ancestry information when this is available. We explore several ways in which this conditioning can be performed: 1) through structural biasing in the attention mechanism with the use of masks, or 2) by constructing structural embeddings via a Graph Convolutional Network (GCN) (Duvenaud et al., 2015) and injecting the information into each layer of the transformer through conditional layer-normalisation (Peebles & Xie, 2023).

**Structural Biasing (Attention Mask)**  We utilise the partial ancestry matrix $\tilde{\mathbf{T}} \in \{-1, 0, 1\}^{(D+2) \times (D+2)}$ to encourage the attention mechanism to respect the underlying causal structure encoded in the matrix. We define a bias matrix $\mathbf{B}$ based on the relationship between a query node $i$ (the effect) and a key node $j$ (the potential cause) in two modes:

- **Soft Attention Bias:** We introduce a learnable bias to encourage attention towards known ancestors and discourage it towards non-ancestors. The attention scores are modified by adding a bias term $\mathbf{B}_{ij}$:

$$\mathbf{B}_{ij} = \begin{cases} +\beta_{\text{anc}} & \text{if } \tilde{T}_{ij} = 1 \text{ (Known Ancestor)} \\ -\beta_{\text{non-anc}} & \text{if } \tilde{T}_{ij} = -1 \text{ (Known Non-Ancestor)} \\ 0 & \text{if } \tilde{T}_{ij} = 0 \text{ (Unknown)} \end{cases} \tag{7}$$

where $\beta_{\text{anc}}$ and $\beta_{\text{non-anc}}$ are learnable positive scalars. This soft bias encourages the model to respect the causal relationships encoded in the matrix, while retaining the flexibility to attend to other variables and thus robustness to incomplete graph information.

- **Hard Attention Mask (Ablation Baseline):** We also study a variant where we enforce strict sparsity. We set $\mathbf{B}_{ij} = -\infty$ if $\tilde{T}_{ij} = -1$ (masking known non-ancestors), and $\mathbf{B}_{ij} = 0$ otherwise. This strictly prevents variables from attending to nodes known not to be their causes.

In both cases, we implicitly set the diagonal $\tilde{T}_{ii} = 1$ to ensure every variable always attends to itself.

**Graph Encoding (GCN)**  We begin by constructing a graph representation where nodes correspond to the variables in the input sequence. Given the partial ancestry matrix $\tilde{\mathbf{T}} \in \{-1, 0, 1\}^{(D+2) \times (D+2)}$, we first initialise node features $\mathbf{Z}^{(0)} \in \mathbb{R}^{(D+2) \times d_{\text{model}}}$ using learnable role embeddings specific to each node type (feature, treatment, or outcome). These initial features are processed by a Graph Convolutional Network (GCN) encoder operating on $\tilde{\mathbf{T}}$ to produce structure-aware node embeddings $\mathbf{Z}_{\text{graph}}$. These embeddings subsequently modulate each transformer block via Adaptive Layer Normalisation (AdaLN) (Peebles & Xie, 2023).

**Transformer Architecture**   We follow Dhir et al. (2025b) and use a transformer architecture (Vaswani et al., 2023) with factorised attention that alternates between processing interactions between features (intra-sample) and between samples (inter-samples). The input to the transformer are the obtained embeddings $\tilde{\mathbf{H}}^{(0)} \in \mathbb{R}^{(S+N_{\text{sink}}) \times (D+2) \times d_{\text{model}}}$, as well as the attention biases $\mathbf{B}$ and the graph encodings $\mathbf{Z}_{\text{graph}}$ if the GCN mechanism is used.

**1. Feature-wise Attention (Graph-Conditioned)**   This layer operates independently on each sample to capture the dependencies between variables. We employ a standard Multi-Head Self-Attention (MHSA) mechanism augmented with the graph conditioning strategies described previously, such that the output of layer $l$ becomes:

$$\mathbf{H}'^{(l)} = \text{MHSA}\left(\text{AdaLN}(\mathbf{H}^{(l-1)}, \mathbf{Z}_{\text{graph}}), \text{mask} = \mathbf{B}\right) + \mathbf{X} \tag{8}$$

where we introduce the conditioning on the GCN embeddings $\mathbf{Z}_{\text{graph}}$ if they are available, and $\mathbf{B}$ applies the structural soft bias. If no GCN embeddings are provided AdaLN is replaced with a LayerNorm.

**2. Sample-wise Attention (Inductive)**   This layer operates independently on each feature dimension to exchange information across samples. Similarly to Dhir et al. (2025b), we first perform MHSA amongst the observational data, followed by Multi-Head Cross Attention (MHCA) from the queries to the observational data for a reduced computational cost (as opposed to masked MHSA).

**3. Position-wise Feed-Forward Network**   The final sub-layer is a standard two-layer MLP with SwiGLU activation (Shazeer, 2020), applied to every token independently. Similar to the feature attention layer, this MLP is conditioned via AdaLN if the graph embeddings are provided, otherwise we use LayerNorm:

$$\mathbf{H}^{(l)} = \text{MLP}(\text{AdaLN}(\mathbf{H}'^{(l)}, \mathbf{Z}_{\text{graph}})) + \mathbf{H}'^{(l)} \tag{9}$$

**Decoding**   After processing through $L$ layers, we obtain the final latent representation $\mathbf{H}^{(L)} \in \mathbb{R}^{(S+N_{\text{sink}}) \times (D+2) \times d_{\text{model}}}$. To produce a prediction for the interventional query, we slice this tensor to extract the representation corresponding to the outcome node of the interventional sample, denoted by $\mathbf{h}_{\text{intv}} \in \mathbb{R}^{d_{\text{model}}}$. Finally, we project $\mathbf{h}_{\text{intv}}$ through a linear prediction head (Linear : $\mathbb{R}^{d_{\text{model}}} \to \mathbb{R}^{K}$) to obtain the parameters of the interventional outcome distribution.

**Bar Distribution**   Following Hollmann et al. (2022), we parameterise the interventional outcome distribution using the bar distribution. This approach partitions the outcome space into $K$ bins with supports $(a_0, a_1], (a_1, a_2], \ldots, (a_{K-1}, a_K]$. We fix $K = 1000$ during training and use equally sized bins. Standardising the outcome variable $y$ both during training and inference ensures a bounded co-domain.

For each interventional query, the model outputs a vector of logits $\boldsymbol{\lambda} \in \mathbb{R}^{K}$. Applying a softmax transformation yields a categorical distribution over bins, $\boldsymbol{\Delta} = \text{softmax}(\boldsymbol{\lambda})$. The resulting bar distribution is given by

$$q_\theta(y \mid \boldsymbol{\lambda}) = \sum_{k=0}^{K-1} \mathbb{I}(a_k < y \leq a_{k+1}) \, \Delta_k. \tag{10}$$

The model is trained using a cross-entropy loss with respect to the bin index of the ground-truth outcome.

# E. Implementation Details

## E.1. The Causal TFM Prior

Our causal prior builds on the principles introduced in DoPFN (Robertson et al., 2025). While DoPFN employs relatively simple structural mechanisms, consisting of a linear function followed by a nonlinearity and additive Gaussian noise, we substantially increase the expressiveness of the mechanisms, inspired by the construction proposed in (Qu et al., 2025). The goal is to expose the model to a broad range of nonlinear, non-Gaussian, and partially discrete data-generating processes.

Sampling a dataset from the prior proceeds in two stages. First, a structural causal model (SCM) $\psi$ is sampled. Second, data are generated from this SCM. Throughout this section, a "dataset" refers to a single sample generated from one SCM; the set of datasets (thus a meta-dataset) of our PFN consists of many such datasets.

### E.1.1. Sampling the SCM

As described in Section 2, an SCM consists of three components: a directed acyclic graph (DAG), a collection of structural mechanisms, and noise distributions. These components are sampled independently, but with several dataset-wide hyperparameters shared across nodes.

**Sampling the DAG** We first sample the number of nodes in the DAG uniformly from $\{2, \ldots, 52\}$, corresponding to the maximum number of nodes considered in our experiments. Next, a dataset-wide edge probability $p$ is drawn from a $\mathrm{Beta}(2, 3)$ distribution. Conditioned on $p$ and the number of nodes, the DAG is sampled using an Erdős–Rényi model with edge probability $p$.

**Sampling the Mechanisms** For each node, the structural mechanism determining its value as a function of its parents is sampled independently. Each mechanism is either implemented as an MLP-based mechanism or as an XGBoost-based mechanism (Chen, 2016). MLP mechanisms yield continuous-valued variables, while XGBoost mechanisms can produce categorical features.

First, a dataset-wide probability of using XGBoost mechanisms is sampled from a categorical distribution with support $\{0.0, 0.1, 0.2, 0.3\}$ and corresponding probabilities $(0.8902, 0.0989, 0.00989, 0.000989)$. For each node, this probability is then used to decide uniformly at random whether its mechanism is implemented as an MLP or as an XGBoost regressor.

MLP mechanisms are implemented using a standard PyTorch (Paszke et al., 2019) multilayer perceptron with default weight initialization. The number of hidden layers is sampled from a categorical distribution with choices $\{0, 1, 2, 3\}$ and probabilities $(0.875, 0.1, 0.025, 0.01)$. The hidden layer width is sampled independently from a categorical distribution with choices $\{1, 2, 4, 6, 8, 10, 12, 14, 16, 32\}$ and corresponding probabilities proportional to $(0.7, 0.2, 0.1, 0.05, 0.04, 0.03, 0.02, 0.01, 0.01, 0.01)$.

The nonlinearities used in the MLP mechanisms follow those introduced in TabICL (Qu et al., 2025). They include standard activation functions such as ReLU, ReLU6 (Krizhevsky et al., 2012), SELU (Klambauer et al., 2017), SiLU (Hendrycks, 2016), Softplus, the hard hyperbolic tangent defined as $\mathrm{HardTanh}(x) = \max(-1, \min(1, x))$, the signum function, the sine function, a radial basis function given by $\mathrm{RBF}(x) = \exp(-x^2)$, and the exponential function. In addition, they include the absolute value $f(x) = |x|$, a clipped indicator-like function $f(x) = \mathbf{1}_{|x| \leq 1}$, and a quadratic nonlinearity $f(x) = x^2$.

Finally, an approximate sample from a Gaussian process of the form $f(x) = \phi(x)^\top z$ is used, where $z \sim \mathcal{N}(0, I)$. The feature map $\phi(x) \in \mathbb{R}^N$ is defined as

$$\phi(x) := \frac{w \odot \sin(ax + b)}{\|w\|_2},$$

with feature dimension $N := 256$. The parameters are sampled independently as $b_i \sim \mathcal{U}[0, 2\pi]$, $a_i \sim \mathcal{U}[0, N]$, and $w_i := a_i^{-\exp(u_i)}$, where $u_i$ is sampled from a standard uniform distribution.

XGBoost mechanisms are implemented as XGBoost regressors (Chen, 2016) fitted on 10 to 500 samples of standard-normal noise for both inputs and targets, analogous to TabICL (Qu et al., 2025). With probability $50\%$, we add standard-normal noise to the output of the regressor. This results in features that are clustered around specific values while not being strictly categorical. The output of the XGBoost regressor is subsequently passed through a nonlinearity sampled in the same manner as for the MLP mechanisms.

Each mechanism additionally takes a noise variable as input. This noise is either added after applying the nonlinearity, added before applying it, or treated as an additional input dimension to the mechanism and thus mixed into the input vector. For variables without parents, we sample a value from the exogenous noise distribution (described below) and pass it through a mechanism sampled as described above. Furthermore, with a probability of $50\%$, we standardise (using BatchNorm) the output of each mechanism.

**Sampling the Noise Distributions**  To sample the noise distributions in the SCM, we distinguish between exogenous noise, which determines the values of variables without parents, and endogenous noise, which together with the parent values determines the values of variables with parents. Both noise distributions are constructed in the same way, but with different dataset-wide hyperparameters.

Specifically, the noise distribution is defined as an equal-weight mixture of Normal, Gamma, Laplace, Student-$t$, and Gumbel distributions, with zero mean. The standard deviation is sampled from a Gamma distribution $\mathrm{Gamma}(\mu, \sigma)$ with mean $\mu$ and standard deviation $\sigma$. For exogenous noise, $\mu$ is sampled from a $\mathrm{LogNormal}(1, 1)$ distribution and $\sigma$ from a $\mathrm{Uniform}(0.1, 0.4)$ distribution. For endogenous noise, $\mu$ is sampled from a $\mathrm{LogNormal}(-3, 6)$ distribution and $\sigma$ from a $\mathrm{Uniform}(0.0, 0.5)$ distribution. All noise hyperparameters are fixed within a dataset.

### E.1.2. SAMPLING DATA

We use the sampled SCMs to generate both purely observational data and paired observational–interventional data.

**Observational Data**  To obtain an observational dataset from an SCM $\psi$, we apply ancestral sampling as described in Section 2. The number of training points $N$ is sampled uniformly between 1 and 1000, with the remaining $1000 - N$ points used as test data. Ancestral sampling yields a table of training and testing data, from which one column is uniformly at random designated as the target and the remaining columns as input features.

All test features are standardised by subtracting the empirical mean and dividing by the empirical standard deviation. The target variable is normalised to lie in the interval $[-1, 1]$. Outliers are removed by clipping values above the 99.5th percentile $q_{99.5}$ to $q_{99.5}$ and values below the 0.5th percentile $q_{0.005}$ to $q_{0.005}$.

To avoid implausible datasets during training, we reject datasets whose target variance after normalisation is smaller than $0.01$ or that exhibit fewer than $20\%$ unique target values across training samples.

**Observational–Interventional Data**  To sample paired observational and interventional datasets from an SCM $\psi$, we first uniformly at random assign two distinct nodes in the causal graph to serve as the treatment variable $t$ and the outcome variable $y$. An observational training dataset is then obtained via ancestral sampling.

Importantly, to simulate interventions, we perform a hard intervention on a particular node $t$ by removing all its incoming edges. Subsequently, we simulate an interventional test-dataset by drawing i.i.d. samples from the intervened-upon SCM. A critical choice here is the distribution $p_{int}(t)$ from which to sample the values of $T$ that are used to perform the intervention. (This distribution is not prescribed via the SCM itself). We choose $p_{int}(t)$ to be the same distribution as the marginal distribution over $T$ under the observational distribution induced via the SCM $\psi$, i.e.,

$$p_{int}(t|\psi) = \int p(t, \mathbf{x}, y|\psi) d\mathbf{x} dy. \tag{11}$$

This means that the distribution of interventional values depends on the distribution of observational data within a specific SCM—and, critically, matches exactly observational distribution of treatments. We implement this by performing two passes through the not-intervened-upon SCM: The first one samples the observational dataset; for the second forward pass we sample different noise values, but keep everything the same, and only collect the values for the treatment variable. Then, the SCM is intervened upon, we resample the noise again, and use the observational treatment values from the previous pass to simulate the interventions.

The paired observational–interventional data are preprocessed in the same manner as the purely observational data, including standardisation of features, normalisation of the target, outlier removal, and rejection of implausible datasets.

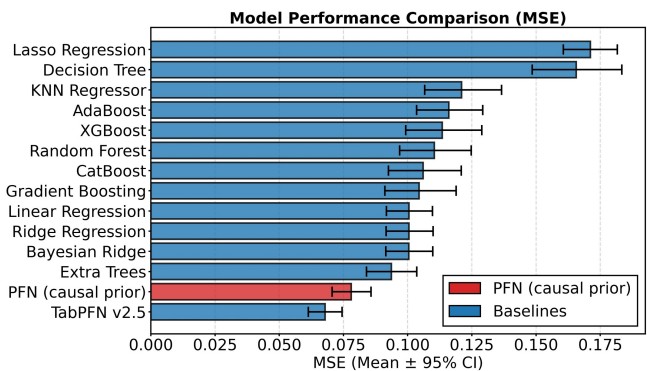

*Figure 5.* Predictive performance on small-scale datasets of a PFN trained on our causal prior compared to standard baselines. We subsample datasets with up to 1000 training points from the TabArena regression benchmark and consider out-of-the-box performance, i.e., default hyperparameters. A PFN trained on our prior achieves strong performance compared to the baselines, but is outperformed by TabPFN v2.5.

### E.2. Empirically Validating the Causal Prior

To the best of our knowledge, all existing priors for tabular foundation models are not natively causal and do not directly support the generation of interventional data (Hollmann et al., 2022; Qu et al., 2025; Zhang et al., 2025a; Qu et al., 2026). The priors for state-of-the-art commercial models such as TabPFN (Hollmann et al., 2025) or LIMIX (Zhang et al., 2025b) might be implemented as SCMs; however, this information is neither publicly disclosed nor is the code accessible. The priors by Robertson et al. (2025); Dhir et al. (2025b); Ma et al. (2025) are causal in their implementation, but are rather simple (they do not allow sampling categorical variables, for instance) and only evaluate on small-scale synthetic or semi-synthetic causal benchmarks.

We therefore developed a custom, *natively causal*, TabPFN-style prior that allows to generate observational, as well as interventional data while giving access to the SCMs used to generate the data. This prior builds upon the simple causal prior of Robertson et al. (2025) and uses nonlinearities inspired by TabICL (Qu et al., 2025). More specifically, our prior allows to sample SCMs by (i) drawing the number of nodes uniformly between 2 and 52, (ii) sampling a causal graph via the Erdős–Rényi scheme. Subsequently, (iii) the distribution over exogenous and endogenous variables is randomly determined, followed by (iv) sampling functional mechanisms as randomly initialised Multi-Layer Perceptrons (MLPs) or as XGboost regressors (Chen, 2016) fitted to Gaussian noise. An SCM from the prior can then be used to directly draw observational data, or to sample interventional data after modifying the SCM. We discuss the details in Appendix E.1.

To validate the proposed prior, we train a tabular foundation model in the purely predictive setup and compare its performance on small-scale tabular regression datasets with up to 1000 training samples (as done for the first version of TabPFN). More specifically, we evaluate a model trained on our causal prior using the 13 real-world regression datasets from the TabArena benchmark (Erickson et al., 2025) and compare it against standard tabular baselines. We subsample small-scale datasets with 1000 training and 1000 testing-points 25 times from each of the 13 real-world datasets. All methods use default hyperparameters and our PFN (unlike TabPFN v2.5), uses simple preprocessing and no self-ensembling strategies during inference. Additionally, our PFN is only trained on 3.2 million synthetic datasets compared to 130 million for TabPFN v2 (Hollmann et al., 2025), and probably even more for TabPFN v2.5.

Our experiments show that, on small-scale datasets, the PFN trained on the proposed causal prior achieves similar or slightly better results than untuned baselines while TabPFN v2.5 achieves the lowest mean MSE. (Please see Figure 5).

### E.3. Causal Effects under the Tabular Foundation Model Prior

While a causal prior, validated against real-world data in the observational setting and equipped with the intervention distributions described above, it is of interest to investigate the causal implications of such a TFM prior. Concretely, we want to investigate the (expected) discrepancy $D_c(\psi)$ between the interventional distribution $p(y \mid \mathrm{do}(t), \psi)$ and the observational conditional $p(y \mid t, \psi)$ distributions:

$$D_c(\psi) := \mathbb{E}_{p_{int}(t \mid \psi)} \left[ d(p(y \mid \mathrm{do}(t), \psi), \, p(y \mid t, \psi)) \right], \tag{12}$$

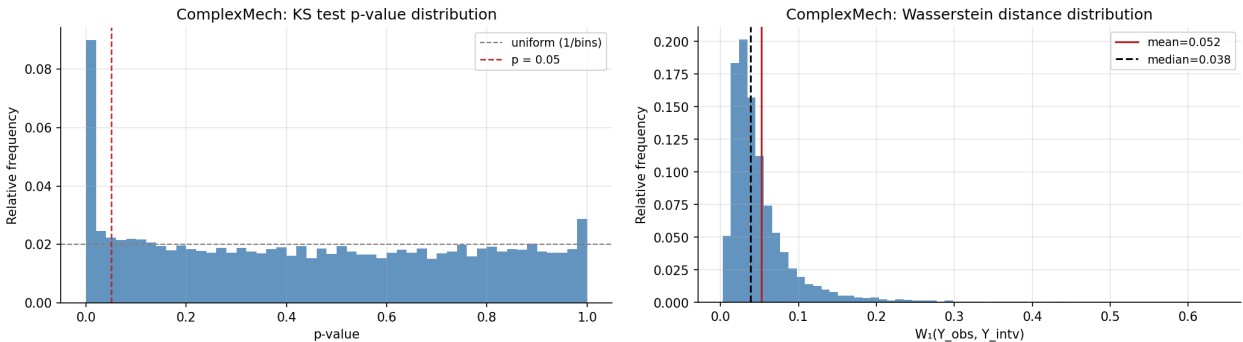

*Figure 6.* Distribution of Kolmogorov–Smirnov (KS) test $p$-values (left) and Wasserstein-1 ($W_1$) distances (right) across 10,000 SCMs sampled from the complex-mechanism prior. Each comparison is between $p(y \mid \mathrm{do}(t), \psi)$ and $p(y \mid t, \psi)$. Small $p$-values indicate detectable discrepancies. While $12.64\%$ of SCMs yield $p < 0.05$, most exhibit weak effects. The $W_1$ distances have mean $0.052$ and show a heavy right tail.

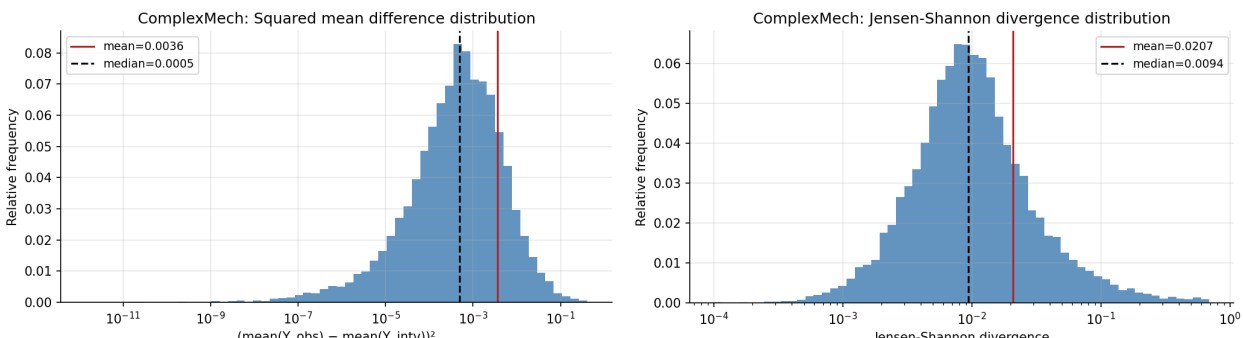

*Figure 7.* Distribution of squared mean differences (left) and Jensen–Shannon (JS) divergences (right) between $p(y \mid \mathrm{do}(t), \psi)$ and $p(y \mid t, \psi)$ across 10,000 SCMs. Both axes are shown on a logarithmic scale. The mean squared difference is $0.0036$ and the mean JS divergence is $0.0207$. Both metrics exhibit heavy-tailed behaviour, indicating that while most causal effects are small, larger effects occur with non-negligible probability.

where $d$ denotes a metric or divergence on probability distributions and $p_{int}(t \mid \psi)$ is the intervention distribution (cf. Appendix E.1.2). We are interested in the distribution of $D_c(\psi)$ for SCMs $\psi \sim p(\psi)$ drawn from the TFM-Prior and the LinGaus-Prior.

To estimate this distribution, we sample 10,000 SCMs from the prior. For each SCM, we compare $p(y \mid \mathrm{do}(t), \psi)$ and $p(y \mid t, \psi)$ using multiple discrepancy measures. First, we compute the $p$-value of a Kolmogorov–Smirnov (KS) test (Massey Jr, 1951) with the null hypothesis that both distributions are identical. This provides an interpretable proxy for $1 - d(\cdot, \cdot)$: small $p$-values indicate that the null hypothesis of equal distributions can be rejected, implying a detectable causal effect.

In addition, we report three direct discrepancy measures: the Wasserstein-1 distance ($W_1$), the squared mean difference, and the Jensen–Shannon (JS) divergence. The squared mean difference is simply the distribution of the squared difference between the mean of the observational and the interventional samples.

Figure 6 shows the distribution of KS $p$-values together with the corresponding $W_1$ distances. While $12.64\%$ of SCMs yield $p < 0.05$, with a pronounced peak for $p$-values below $0.01$, indicating detectable differences between $p(y \mid \mathrm{do}(t), \psi)$ and $p(y \mid t, \psi)$, the majority of datasets exhibit large $p$-values. This suggests that detectable causal effects are relatively rare, but existent, under the prior. The $W_1$ distances have a mean of $0.052$ indicating that most datasets have very small absolute differences between the observational and interventional datasets under the TFM prior. There is, however a relatively heavy tail towards higher $W_1$ distances.

Figure 7 further quantifies these effects using the squared mean difference (mean $0.0036$) and the JS divergence (mean $0.0207$). Both quantities are shown on a logarithmic scale and likewise exhibit heavy-tailed behaviour, reinforcing the

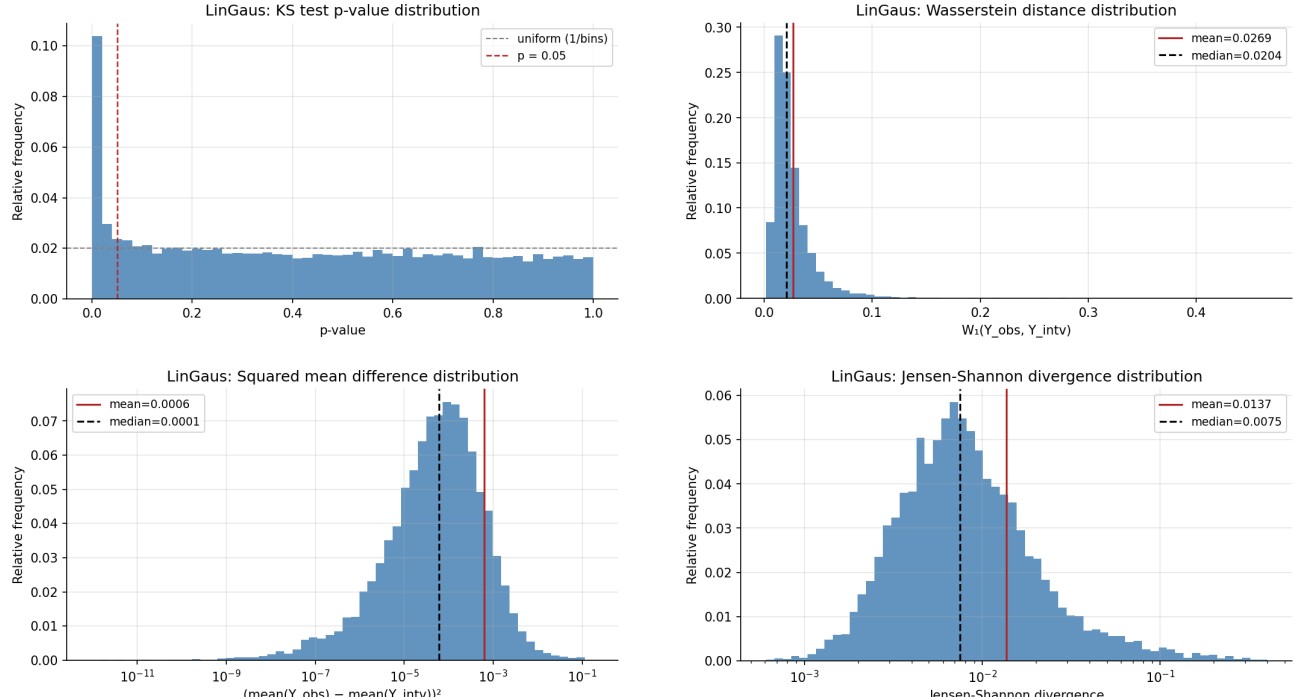

*Figure 8.* Distribution of Kolmogorov–Smirnov test $p$-values (upper left), Wasserstein-1 ($W_1$) distance, squared mean difference and Jensen-Shannon (JS) divergences between $p(y|\text{do}(t),\psi)$ and $p(y|t,\psi)$ on 10,000 SCMs from the linear-Gaussian prior.

observation that the prior predominantly generates weak effects, with occasional larger deviations.

Figure 8 shows that the situation is analogous for the LinGaus prior, which is identical to the TFM prior, except that the noise variables are always Gaussian and the mechanisms are exclusively linear: On 10,000 SCMs from this prior, $14.74\%$ have a detectable difference between the observational and interventional distributions, whereas the absolute effect sizes are, on average, even smaller than in the TFM Prior case.

In summary, using our causal implementation of a TFM prior, together with a natural notion of interventions, leads to detectable, albeit somewhat rare, and, overall, in absolute terms, small differences between the causal interventional distribution $p(y|\text{do}(t),\psi)$ and the observational PPD $p(y|t,\psi)$.

This also leads to the expectation that the absolute size of the effect of graph-conditioning remains rather small, since graphical information primarily matters for causal tasks, and one can expect the effect of utilising graph information to be roughly of the same order of magnitude as the difference between the observational and interventional distributions. The latter is because, in the purely unidentifiable case, one would expect the graph information to move the model from predicting $p(y|t,\psi)$ to predicting $p(y|\text{do}(t),\psi)$.

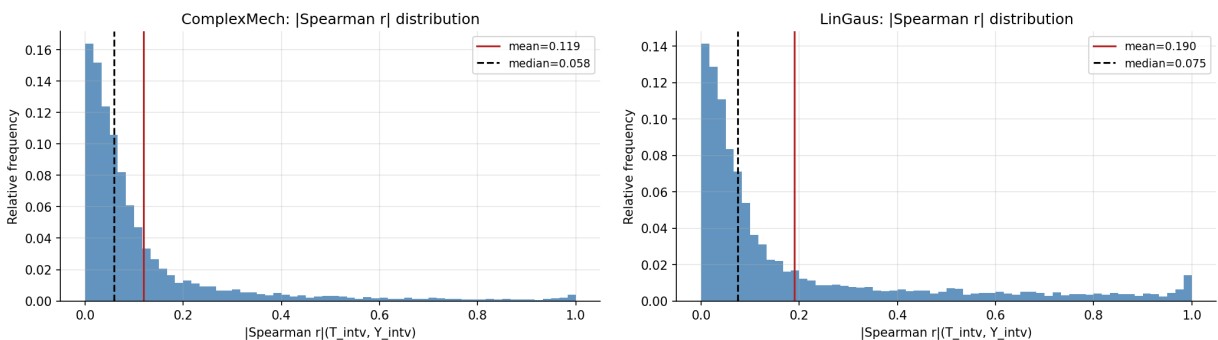

*Figure 9.* Distribution of absolute Spearman correlations between treatment $T$ and outcome $Y$ across 10,000 SCMs sampled from the TFM prior (left) and the linear-Gaussian prior (right). In both cases, most datasets exhibit weak correlations, while a small fraction shows strong monotonic relationships, reflected in a heavy right tail.

### E.3.1. CORRELATION OF TREATMENT AND OUTCOME

Additionally, Figure 9 shows the distribution of the Spearman correlation between the intervention variable $T$ and the outcome $Y$ for 10,000 SCMs that are sampled from the TFM-Prior (on the left) or the LinGaus-Prior (on the right). While in both cases, more than half of the datasets have Spearman correlations below $0.05$, there is a notable heavy tail towards values of $1.0$. More specifically, on the TFM-Prior, $2.6\%$ of datasets have a Spearman correlation of more than $0.7$, and on the LinGaus-Prior $7.9\%$.

This provides further indication of the existence of only a few datasets where a strong relationship exists between the treatment and outcome variables.

# F. Detailed Experimental Setup

## F.1. Which Graph-Conditioning Method Works Best?

To determine the most effective way of conditioning on graphical information in a causal TabPFN, we evaluate different graph-conditioning methods on synthetic case studies in which the causal graph is not identifiable from observational data alone. Specifically, we consider 30,000 datasets sampled from SCMs with linear mechanisms and Gaussian noise, together with internal standardisation (Ormaniec et al., 2025). The datasets contain uniformly distributed 2, 5, 20, 35 and 50 nodes and are drawn in equal proportions from SCMs with (i) a directed path from treatment $T$ to outcome $Y$, (ii) a directed path from $Y$ to $T$, and (iii) no directed path between $T$ and $Y$. All those datasets contain 500 train and test-samples.

The linear mechanisms are implemented as one-layer PyTorch (Paszke et al., 2019) MLPs with default weight initialisation. The standard deviation is sampled from a Gamma distribution Gamma$(\mu, \sigma)$ with mean $\mu$ and standard deviation $\sigma$. For exogenous noise, $\mu$ is sampled from a LogNormal$(1, 1)$ distribution and $\sigma$ from a Uniform$(0.1, 0.4)$ distribution. For endogenous noise, $\mu$ is sampled from a LogNormal$(-3, 6)$ distribution and $\sigma$ from a Uniform$(0.0, 0.5)$ distribution. All noise hyperparameters are fixed within a dataset.

We implement the baseline with an identical architecture compared to the graph-conditioning methods and report the difference from this baseline in terms of negative log-likelihood, mean squared error (MSE), and $R^2$ on the test-data.

## F.2. RealCause Benchmark

We provide results showing the benefit of partial graph conditioning on the semi-synthetic RealCause benchmark (Neal et al., 2020), which transforms real-world Randomized Controlled Trial (RCT) data into unconfounded scenarios by generating synthetic confounding effects as well both potential outcomes $y^0$ and $y^1$. RealCause provides synthetic ground truth values for Conditional Average Treatment Effect (CATE) and Average Treatment Effect (ATE) estimation and is a commonly used benchmark. We evaluate the performance of our graph conditioned model against the same model with no graph information, as well as a model trained on the same prior for the predictive task.

### F.2.1. METRICS FOR REALCAUSE

**Precision in Estimation of Heterogeneous Effects**   We measure performance in estimating Conditional Average Treatment Effects (CATEs) using the root Precision in Estimation of Heterogeneous Effects (PEHE) metric as proposed in (Shalit et al., 2017), defined as

$$\sqrt{\text{PEHE}} = \sqrt{\frac{1}{n} \sum_{i=1}^{n} \left( \hat{\tau}(x_i) - \tau(x_i) \right)^2}$$

Here, $\hat{\tau}(x) = \int q_\theta(y|\text{do}(1), \boldsymbol{x}, \mathcal{D}) - q_\theta(y|\text{do}(0), \boldsymbol{x}, \mathcal{D}) dy$ is the CATE estimated by the model for a binary treatment variable $t \in \{0, 1\}$, and $\hat{\tau}(x_i) = y_i^1 - y_i^0$, for $y_i^1$ and $y_i^0$ the two potential outcomes for individual $i$.

**Relative Average Treatment Effect Error**   For Average Treatment Effects (ATEs), we report the *relative ATE error*

$$\epsilon_{\text{ATE}} = \frac{\left| \widehat{\text{ATE}} - \text{ATE} \right|}{|\text{ATE}|},$$

where the true ATE is

$$\text{ATE} = \frac{1}{n} \sum_{i=1}^{n} \tau(x_i) = \frac{1}{n} \sum_{i=1}^{n} \left( y_i^1 - y_i^0 \right),$$

and the estimated ATE is

$$\widehat{\text{ATE}} = \frac{1}{n} \sum_{i=1}^{n} \hat{\tau}(x_i),$$

with $\hat{\tau}(x)$ denoting the model's CATE estimate defined above.

### F.3. Results on the Balanced PSID Dataset

On the PSID dataset used in RealCause (Neal et al., 2020), we find that the predictive model achieves a relative error close to one, i.e., performance close to that of a naive constant baseline (Table 1). The same applies to the ancestral-conditioned model as well as the non-ancestral-conditioned model. To remove the effect of this imbalance when investigating the effect of graph-conditioning in terms of the number of treated (141 individuals) versus untreated (2266 individuals), we perform rebalancing by randomly subsampling 500 individuals from the untreated group while keeping all treated individuals. We find that this improves the performance of all methods, showing again a pronounced benefit of performing graph-conditioning in Table 2.

*Table 2.* Results on the **PSID (balanced)** semi-synthetic dataset. We report root Precision in Estimation of Heterogeneous Effects ($\sqrt{\text{PEHE}}$) and relative ATE error ($\epsilon_{\text{ATE}}$). Mean $\pm$ standard deviation over 100 runs.

| Method | $\sqrt{\text{PEHE}}$ | $\epsilon_{\text{ATE}}$ |
|---|---|---|
| Predictive | $22045\pm136$ | $0.95\pm0.01$ |
| No Ancestral Information | $21896\pm137$ | $0.94\pm0.00$ |
| Ancestral Information | $\mathbf{19711\pm230}$ | $\mathbf{0.65\pm0.02}$ |

## G. Training Details

We train all models for 50,000 steps with a batch-size of 32. To save memory, we accumulate the gradients from 4 forward-passes with a batch-size of 8 each. This corresponds to training on 1.6 million synthetic datasets. Each of those datasets has a size of 1000 samples that is split uniformly at random between the context- and target-set. Our model has 7,962,910 parameters in the predictive setting, 8,012,012 parameters when doing the Soft Attention biasing and 11,186,463 parameters for the version that combines the GCN with Soft Attention.

We train with the Adam optimiser (Kingma, 2014) at a learning rate of $10^{-4}$, using cosine annealing (Loshchilov & Hutter, 2017) with a warm-up ratio of 0.1 and a minimum learning rate-ratio of 0.1. The weight decay parameter is set to $10^{-5}$, and we do not use dropout. For efficiency, we utilise Flash Attention (Dao et al., 2022) and train in 16-bit precision. All our neural networks and their training are implemented using PyTorch (Paszke et al., 2019). The training of each model takes approximately 2 days and three hours on a single H100 GPU with 80 gigabytes of memory.

## H. Additional Results

### H.1. Additional Results on Linear-Gaussian Data

We provide additional quantitative results for the linear-Gaussian synthetic setting discussed in the main text.

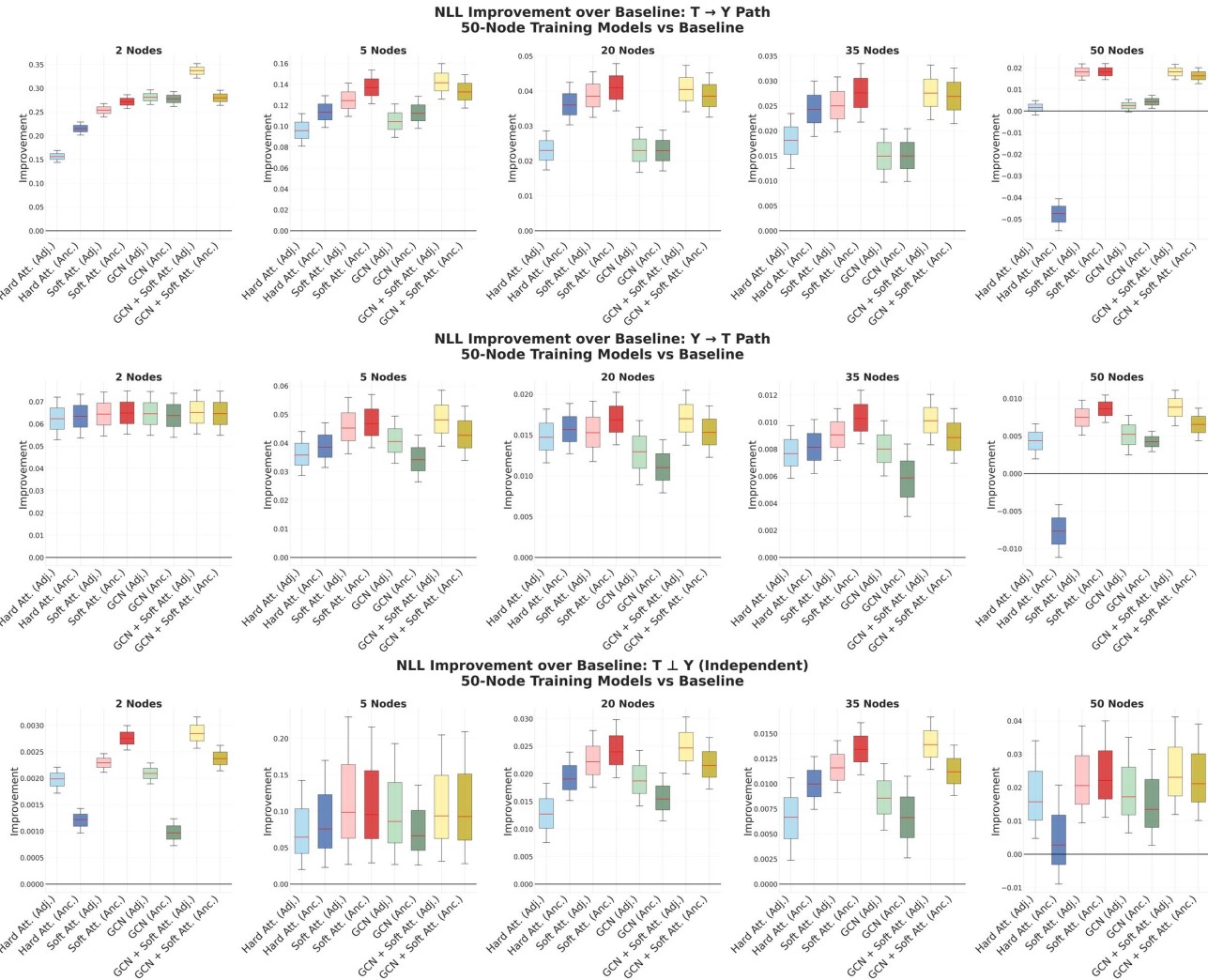

*Figure 10.* Results, measured in terms of negative log-likelihood (NLL) on held-out test data, for the different graph-conditioning methods evaluated on synthetic data generated from SCMs with linear mechanisms and Gaussian noise. Overall, the graph-conditioned models substantially outperform the baseline that does not use graph information. One notable exception occurs in the 50-node setting: using hard attention together with ancestry-based conditioning leads to very poor performance, indicating instability of this approach. Importantly, this failure mode does not appear for smaller numbers of nodes. Apart from this exception, we consistently observe that using soft attention alone, or soft attention combined with a GCN-based conditioning mechanism, yields the best performance across all node counts and across all SCM variants ($T \rightarrow Y$, $Y \rightarrow T$, and $T$ and $Y$ independent).

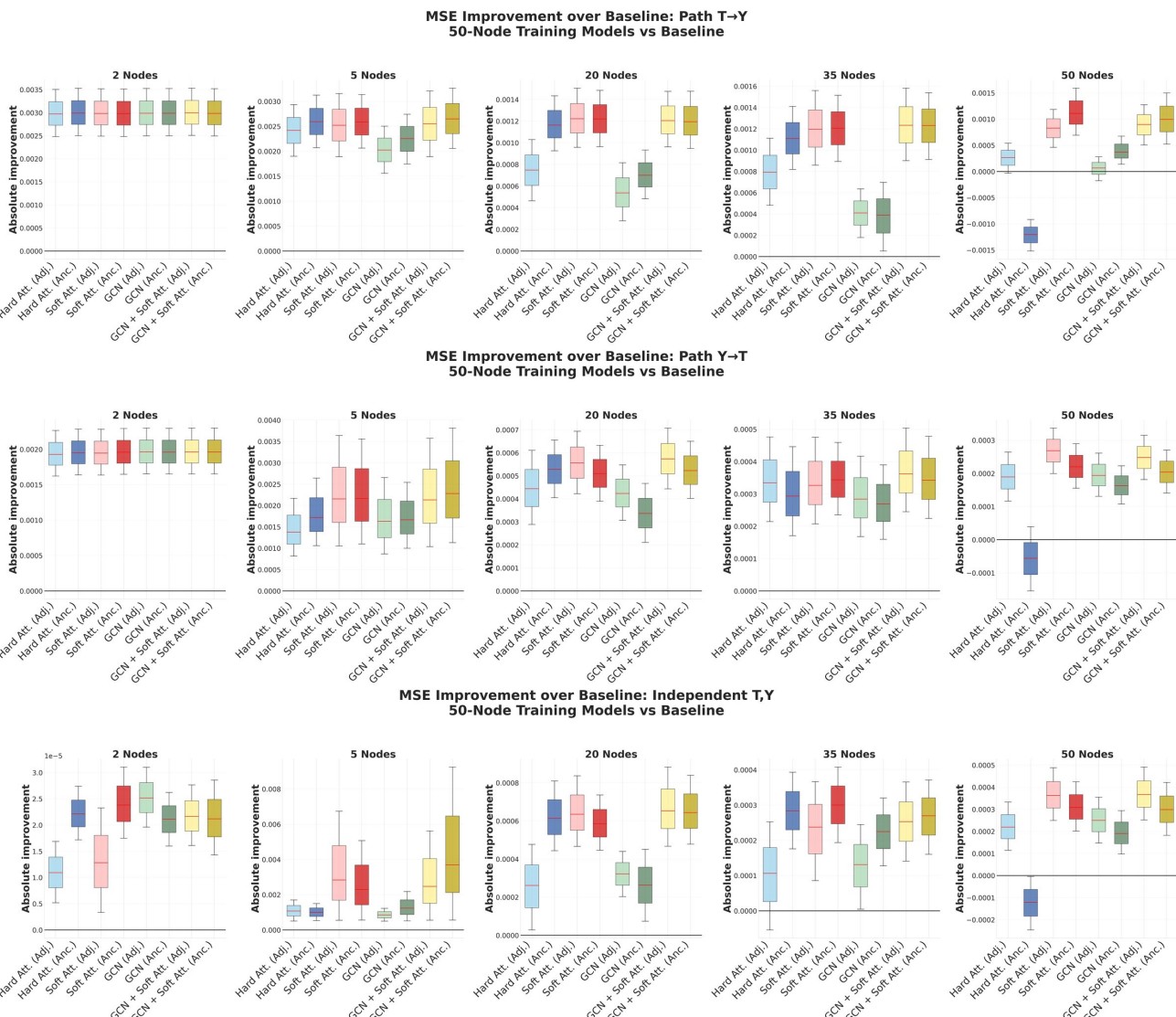

*Figure 11.* Performance comparison of different graph-conditioning methods measured in terms of mean squared error (MSE) on held-out test data. In contrast to negative log-likelihood, which evaluates the quality of the full predictive posterior, MSE assesses only the accuracy of the posterior mean. Overall, the qualitative trends closely mirror those observed for NLL, with graph-conditioned models outperforming the non-graph-conditioned baseline and Soft-attention leading to the best results. However, performance differences between methods are generally smaller under the MSE metric, particularly in the two-node setting.

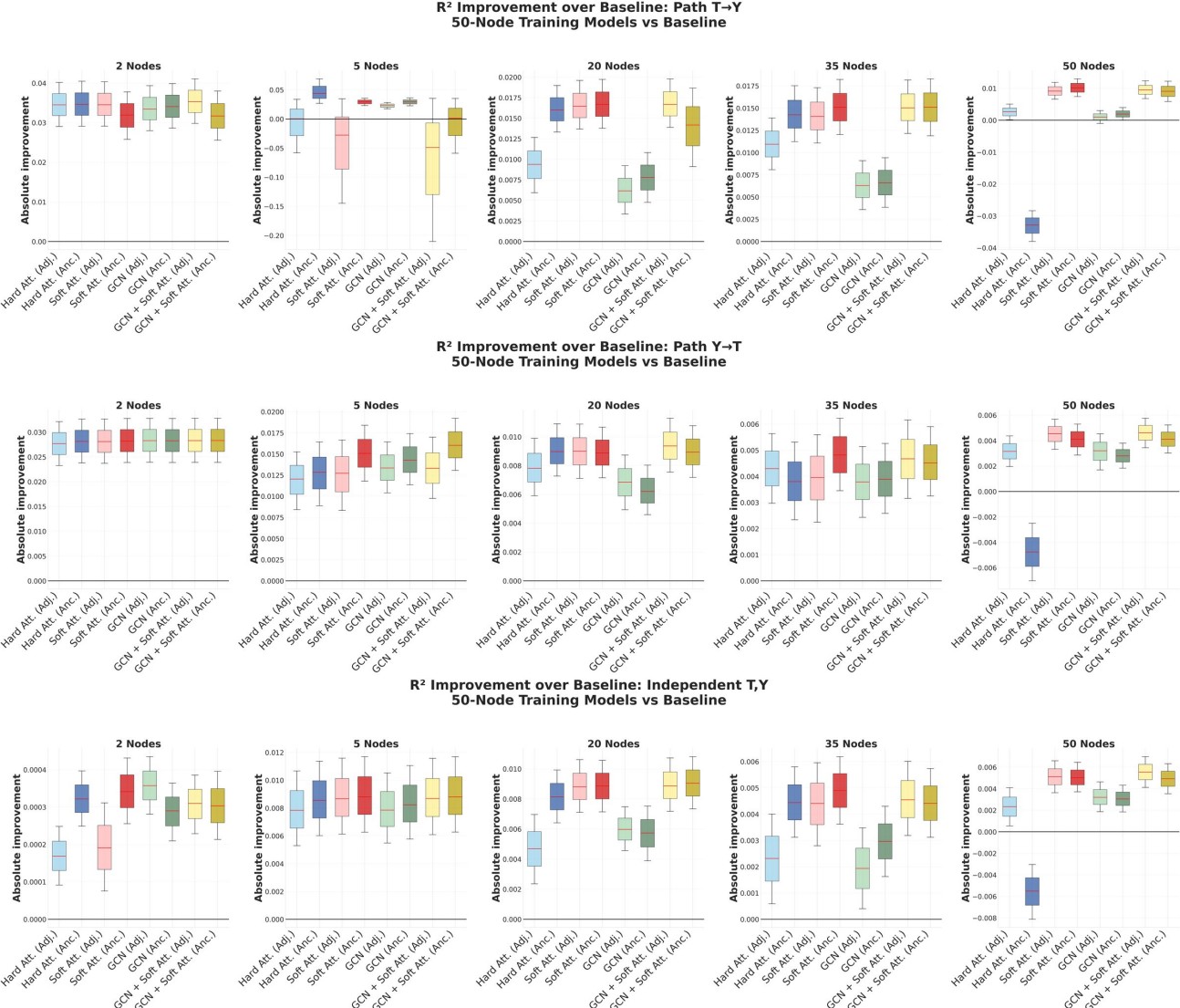

*Figure 12.* Performance comparison of different graph-conditioning methods measured in terms of the coefficient of determination ($R^2$) on held-out test data. Overall, the qualitative behaviour closely matches that observed for the MSE metric, with graph-conditioned models consistently outperforming the non-graph-conditioned baseline and soft-attention–based approaches achieving the strongest performance. One minor exception occurs in the five-node setting, where results for soft attention conditioned on adjacency matrices and for the combined GCN + soft-attention (Adj.) approach exhibit increased variance. This effect is not observed consistently across other node counts.

## H.2. Is graph-conditioning still effective with partial, rather than full ancestry information?

This section provides detailed results for the experiment summarised in Section 4.2. Our goal is to evaluate whether a single model can be trained to operate effectively across varying degrees of available ancestral information.

We consider the same linear-Gaussian setting as detailed in Appendix F.1. Three models are trained under identical conditions, differing only in how structural information is provided:

- **Complete Ancestral Information:** The full ancestor matrix is always available during training.

- **No Graph Conditioning:** No structural information is provided.

- **Amortised (Partial Ancestral Matrix):** During training, a fraction of entries in the ancestor matrix is replaced by $0$, representing unknown relationships. The hide fraction is sampled uniformly, forcing the model to handle different levels of structural knowledge.

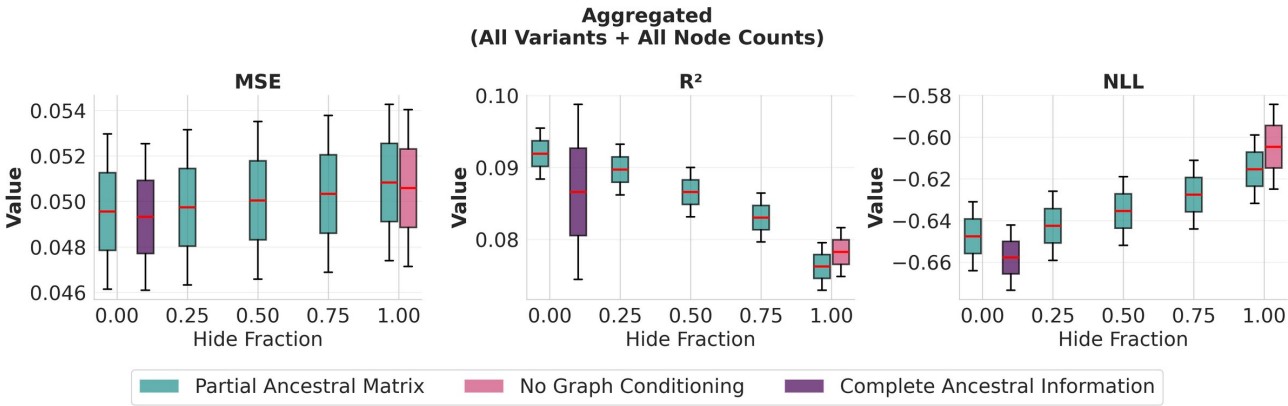

*Figure 13.* Aggregate performance as a function of available structural information. The "Hide Fraction" denotes the proportion of ancestor-matrix entries replaced by $0$ (unknown). Performance degrades monotonically as less information is available.

Figure 13 shows performance aggregated across metrics. As expected, predictive accuracy degrades monotonically as more ancestral entries are hidden, indicating that the model consistently exploits structural information when available.

Crucially, the amortised model remains competitive with both specialised models across all regimes. When no information is hidden, its performance mostly matches that of the model trained exclusively with full ancestral information. When all information is hidden, its performance is similar to that of the model trained without graph conditioning. The differences are rather small and fall within confidence intervals in terms of MSE and $R^2$, suggesting that amortising over information levels does not introduce a substantial performance penalty. For NLL, the fully-informed model has a more pronounced advantage when full information is available, while the amortised model performs marginally better in the fully-hidden regime.

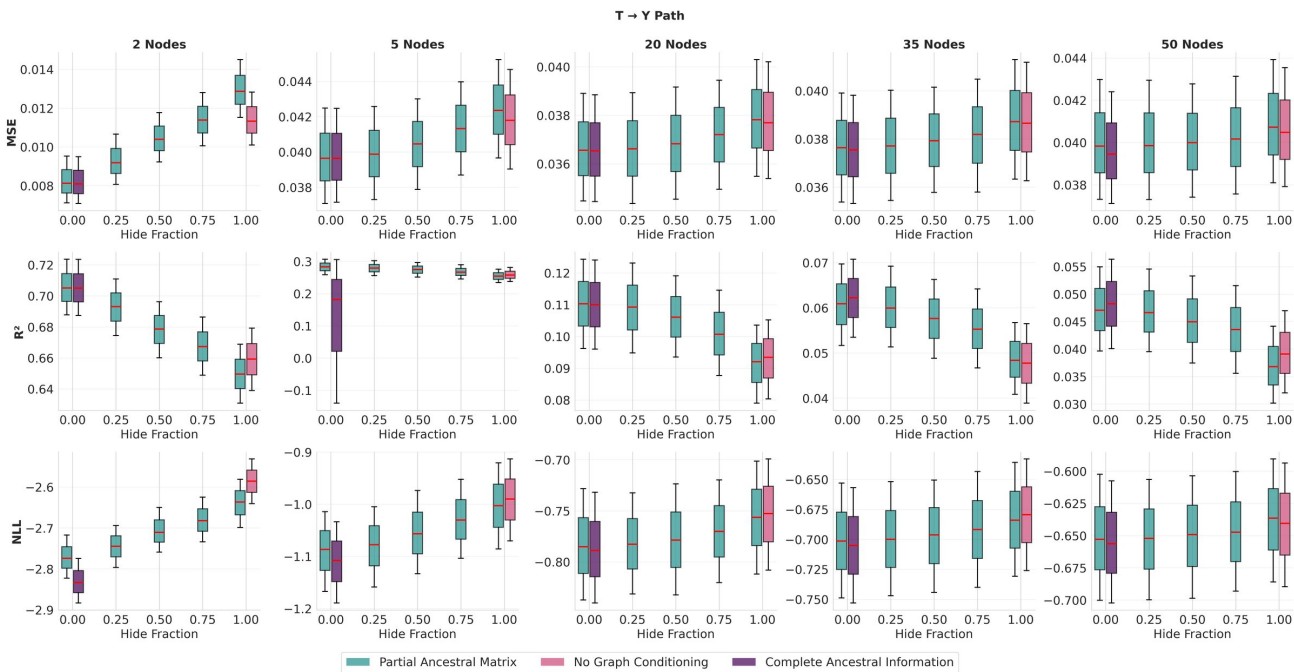

*Figure 14.* Performance across information levels for SCMs with a directed causal path from treatment to outcome ($T \rightarrow Y$).

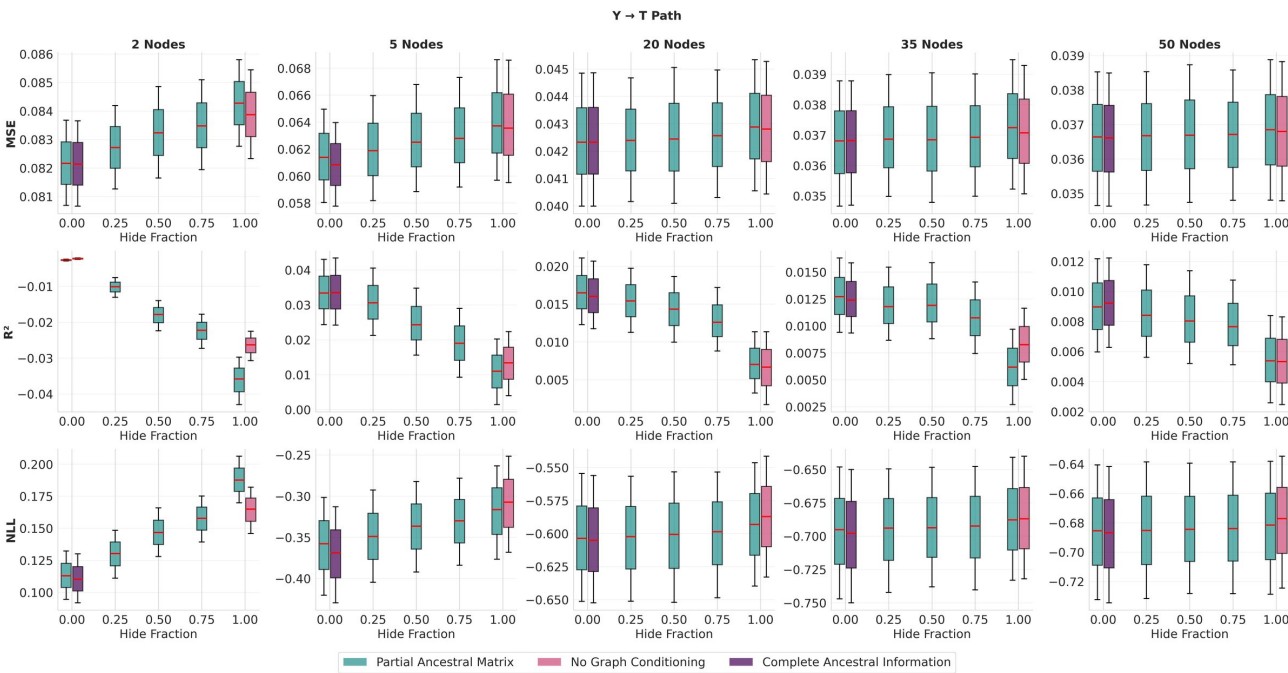

*Figure 15.* Performance across information levels for SCMs with a reverse causal path from outcome to treatment ($Y \rightarrow T$).

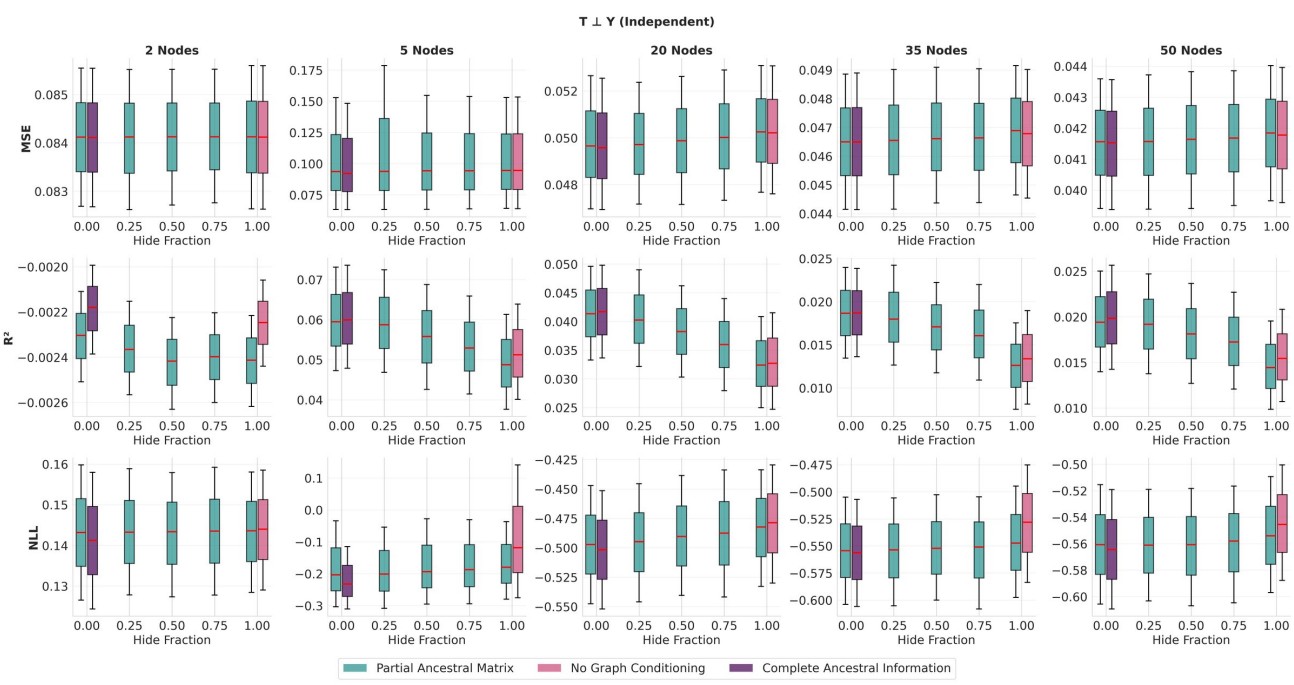

*Figure 16.* Performance across information levels for SCMs where treatment and outcome are causally independent.

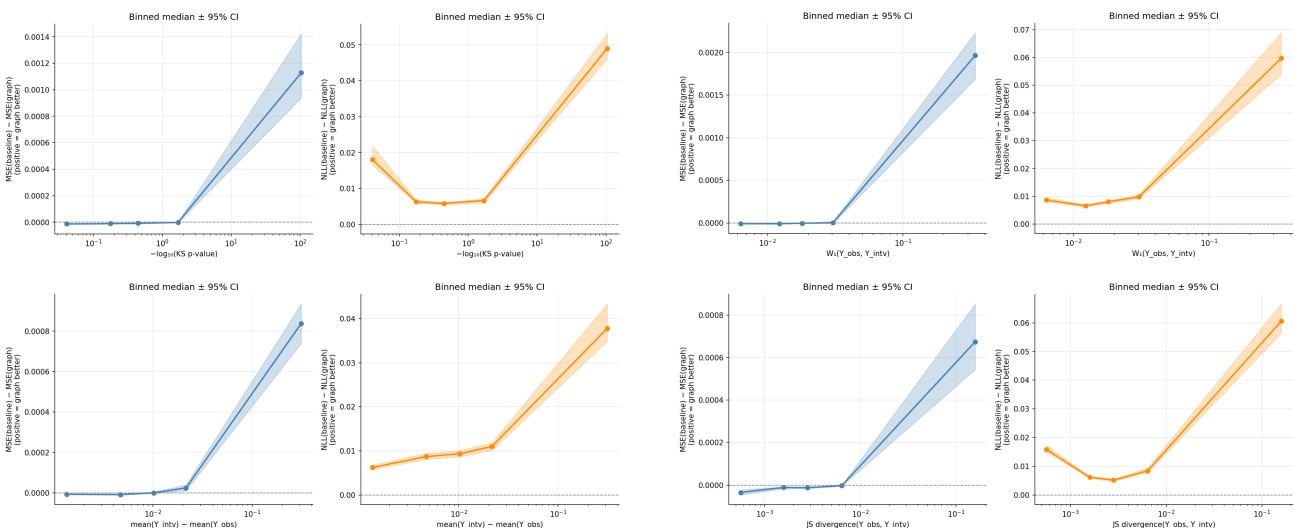

*Figure 17.* Graph-conditioning benefit (MSE and NLL reduction relative to a graph-agnostic baseline, positive values indicate the graph-conditioned model is better) as a function of causal effect strength, measured by four complementary metrics: $-\log_{10}$ of the Kolmogorov–Smirnov (KS) two-sample $p$-value (upper left), Wasserstein-1 distance $W_1$ (upper right), absolute mean shift, and Jensen–Shannon (JS) divergence (lower right), all between the observational outcome distribution $p(y \mid t, \psi)$ and the interventional distribution $p(y \mid \mathrm{do}(t), \psi)$. Each curve shows the binned median over 32,000 ComplexMech samples, binned into five quantile-equal groups along the x-axis (log scale); shaded bands are 95% bootstrap confidence intervals on the median.

# I. Properties of the Prior and the Effect of Graph Conditioning

In this section, we investigate how specific causal properties of our TFM-Prior (see Appendix E.1) relate to the effectiveness of graph-conditioning.

We investigate the effect of graph-conditioning dependent on the difference between the interventional CID $p(y|\mathrm{do}(t), \psi)$ and the observational posterior $p(y|t, \psi)$, as discussed in Appendix E.3. Figure 17 shows that for all metrics, the benefits of graph-conditioning are much larger when the difference between the observational and interventional distribution is above the 80% quantile. In terms of MSE, the difference is basically zero in all metrics except when the difference between observational and interventional distributions is the largest. The difference in negative log-likelihood exhibits, at least in terms of the KS $p$-value, Wasserstein-1 distance and JS-divergence a bimodal effect: for the smallest and largest differences, graph-conditioning helps the most, with a much larger benefit for the largest values. This is arguably because in terms of predicting the whole distribution (which the NLL assess), it is also helpful to know that there is no causal effect in the data.

## J. Comparison to Other Causal Foundation Models

CausalPFN (Balazadeh et al., 2025) and CausalFM (Ma et al., 2025) are trained exclusively for specific identification settings and cannot handle datasets where this criterion is violated. By contrast, our approach is generic: it is trained on a broad prior over causal structures and conditions on graphical information at inference time. This makes a direct performance comparison non-trivial, as the three methods differ both in their priors and in the assumptions they impose. We therefore evaluate all methods on data sampled from our own complex-mechanisms prior, which allows us to control for prior differences and isolate the effect of graph conditioning and identification assumptions.

### J.1. Comparison on the Complex-Mechanisms Prior

Figure 18 reports the mean difference in MSE between our graph-conditioned models and CausalPFN (left panel) and CausalFM (right panel), evaluated on 30,000 datasets from our complex-mechanisms prior. We restrict to datasets with a binary treatment variable to enable fair comparison with CausalPFN and CausalFM. Datasets are stratified into two groups: (a) datasets where all observed covariates form a valid backdoor adjustment set—matching the identification assumption of CausalPFN and CausalFM, which is assumed by both—and (b) datasets where this condition is violated. 95% bootstrap confidence intervals are computed via paired bootstrapping on the same set of results to enable direct uncertainty quantification of the differences.

Our graph-conditioned models (Soft Attention and GCN + Soft Attention) perform only slightly better than CausalPFN when the backdoor criterion is satisfied. When the backdoor criterion is violated—a setting where CausalPFN and CausalFM rely on a misspecified assumption—the advantage of our graph-conditioned models increases substantially: the improvement over CausalPFN is statistically significant at more than 22 standard errors. CausalFM is consistently outperformed by all other methods.

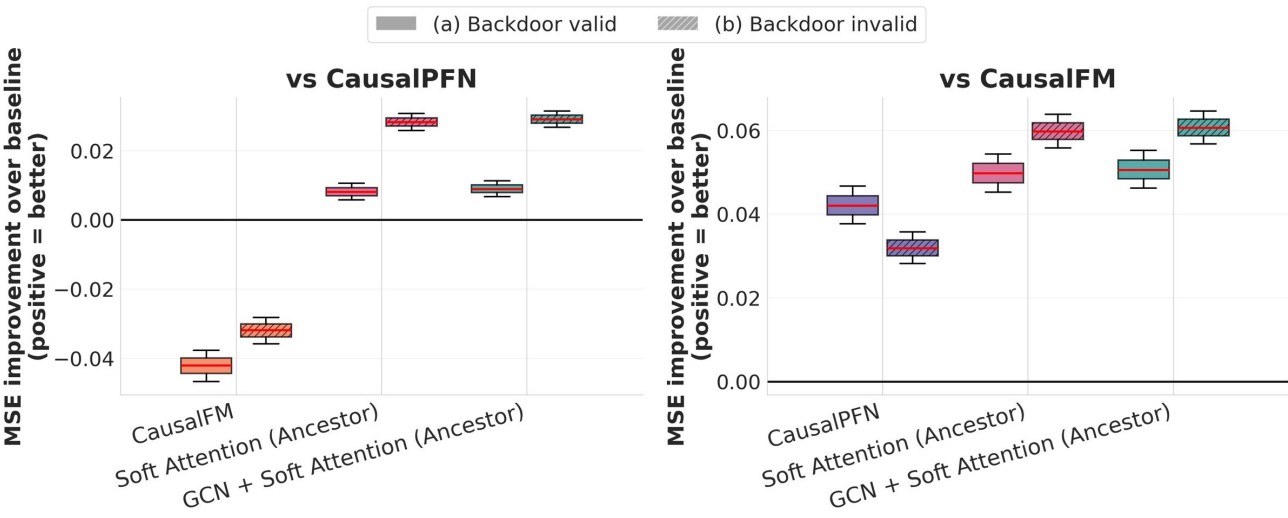

*Figure 18.* Performance under violation of the backdoor criterion on datasets from our complex-mechanisms prior. We compare against CausalPFN (left) and CausalFM (right) using 30,000 datasets with a binary treatment variable, stratified by whether all observed covariates form a valid backdoor adjustment set. We report the mean **difference** in MSE relative to the respective baseline; errors are 95% bootstrap confidence intervals from paired bootstrapping. **Left:** Our graph-conditioned models perform slightly better than CausalPFN when the backdoor criterion is valid; this advantage grows substantially when the criterion is violated (CausalPFN's performance, represented by 0, lies more than 22 standard errors outside the confidence intervals). CausalFM is always significantly worse than CausalPFN. **Right:** All methods outperform CausalFM; the advantage of our graph-conditioned models over CausalFM increases when the backdoor criterion is violated.

*Table 3.* Extended RealCause benchmark results including CausalFM, CausalPFN, and Do-PFN. The first three rows reproduce Table 1 from the main paper. The last three rows show other CFMs, both trained exclusively on graphs satisfying the backdoor criterion. Our graph-conditioned model ("Ancestral Info") consistently outperforms CausalFM and Do-PFN. CausalPFN achieves the best performance overall, which we attribute to its training data closely mimicking the RealCause data-generating process rather than to any advantage in graph conditioning.

| | IHDP | | ACIC | | CPS | | PSID | | PSID (balanced) | |
|---|---|---|---|---|---|---|---|---|---|---|
| | $\sqrt{\text{PEHE}}\downarrow$ | $\epsilon_{\text{ATE}}\downarrow$ | $\sqrt{\text{PEHE}}\downarrow$ | $\epsilon_{\text{ATE}}\downarrow$ | $\sqrt{\text{PEHE}}\downarrow$ | $\epsilon_{\text{ATE}}\downarrow$ | $\sqrt{\text{PEHE}}\downarrow$ | $\epsilon_{\text{ATE}}\downarrow$ | $\sqrt{\text{PEHE}}\downarrow$ | $\epsilon_{\text{ATE}}\downarrow$ |
| Predictive | $6.79 \pm 0.81$ | $0.81 \pm 0.11$ | $3.14 \pm 0.47$ | $0.38 \pm 0.06$ | $11393 \pm 31$ | $0.78 \pm 0.00$ | $\mathbf{11820 \pm 41}$ | $1.03 \pm 0.01$ | $22045 \pm 136$ | $0.95 \pm 0.01$ |
| No Ancestral Info | $6.28 \pm 0.79$ | $0.67 \pm 0.05$ | $3.47 \pm 0.47$ | $0.46 \pm 0.09$ | $12800 \pm 55$ | $0.99 \pm 0.01$ | $13096 \pm 26$ | $\mathbf{0.98 \pm 0.00}$ | $21896 \pm 137$ | $0.94 \pm 0.00$ |
| Ancestral Info | $5.49 \pm 0.78$ | $0.49 \pm 0.08$ | $2.79 \pm 0.45$ | $0.17 \pm 0.08$ | $11213 \pm 60$ | $0.70 \pm 0.02$ | $12975 \pm 24$ | $1.09 \pm 0.01$ | $\mathbf{19711 \pm 230}$ | $\mathbf{0.65 \pm 0.02}$ |
| CausalFM | $5.86 \pm 7.25$ | $0.70 \pm 1.23$ | $3.30 \pm 1.59$ | $0.33 \pm 0.11$ | $12400 \pm 157$ | $0.94 \pm 0.00$ | $22436 \pm 1296$ | $0.95 \pm 0.01$ | $21071 \pm 1339$ | $0.86 \pm 0.05$ |
| CausalPFN | $\mathbf{0.58 \pm 0.07}$ | $\mathbf{0.03 \pm 0.00}$ | $\mathbf{0.92 \pm 0.11}$ | $\mathbf{0.06 \pm 0.01}$ | $\mathbf{8956 \pm 21}$ | $\mathbf{0.14 \pm 0.01}$ | $\mathbf{14402 \pm 198}$ | $\mathbf{0.22 \pm 0.02}$ | $\mathbf{14833 \pm 240}$ | $\mathbf{0.26 \pm 0.02}$ |
| Do-PFN | $6.07 \pm 8.94$ | $0.93 \pm 3.98$ | $4.11 \pm 1.64$ | $0.66 \pm 0.12$ | $12015 \pm 319$ | $0.88 \pm 0.06$ | $20907 \pm 1375$ | $0.93 \pm 0.07$ | $22893 \pm 1606$ | $1.09 \pm 0.12$ |

## J.2. Comparison on the RealCause Benchmark

Table 3 extends Table 1 by including results for CausalFM, CausalPFN, and Do-PFN (Robertson et al., 2025) on all RealCause datasets. CausalPFN performs best overall, which can be attributed to its training process being designed precisely for this case (Neal et al., 2020; Balazadeh et al., 2025): both RealCause and CausalPFN use a DGP with a binary treatment drawn from a neural-network propensity model and potential outcomes generated from a shared parametric family, with treatment heterogeneity controlled in essentially the same way. This prior alignment, rather than the absence of graph conditioning, explains CausalPFN's advantage. Once graph conditioning is introduced, our model outperforms CausalFM and Do-PFN across almost all datasets.

## K. Out-of-Distribution Performance: Misspecification of Graph Information

**Qualitative illustration**  In Figure 19, we demonstrate what happens when providing a CFM with incorrect graph information in the unidentifiable bivariate case of a linear mechanism and Gaussian noise. Since the data does not provide any evidence regarding the causal direction, the model—while performing correct inference under the wrong assumption—outputs what is effectively a "wrong" posterior. This is consistent with traditional causal approaches: if the true causal direction is not identifiable from observations and the stated graph is wrong, the resulting causal estimate will be wrong. Robustness to misspecification is therefore limited in essentially the same way as for classical causal methods.

**Quantitative comparison: hiding vs. misspecifying**  Figure 20 shows a more important finding: a comparison between hiding edges (marking them as "unknown") and misspecifying them (flipping their sign) as a fraction of the total entries of the ancestral matrix is corrupted. While misspecifying edges—providing an incorrect causal direction—leads to a rapid and substantial performance degradation, hiding the same edges degrades performance only gradually. This asymmetry directly motivates partial ancestral matrices as a practical tool: a practitioner who is unsure about a specific causal relationship is much better served by leaving it unspecified than by committing to a direction that might be wrong. This is a key practical advantage that distinguishes our approach from methods that require a fully specified graph.

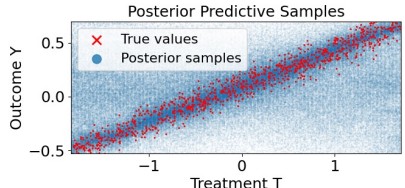 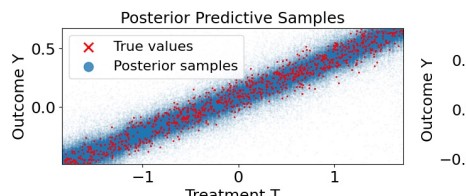

*Figure 19.* Posterior predictive samples in the two-node case where $T$ causes $Y$ ($T \rightarrow Y$); more specifically $Y = 0.25 \cdot T + \epsilon$, for $\epsilon \sim \mathcal{N}(0, 0.1)$. Here, the causal direction is not identifiable from observational data. **Left**: Without telling the model $Q_\theta$ trained on arbitrary mechanisms (as, e.g., in Robertson et al. (2025) or Dhir et al. (2025b)) the correct causal direction, it outputs a mixture distribution $Q_\theta(Y|\mathrm{do}(T), T \overset{?}{\longleftrightarrow} Y) = 0.5 \cdot P(Y|\mathrm{do}(T), Y \rightarrow T) + 0.5 \cdot P(Y|\mathrm{do}(T), T \rightarrow Y)$. **Middle**: When conditioned on the right causal graph, the output aligns with the correct causal effect: $Q_\theta(Y|\mathrm{do}(T), T \rightarrow Y) = P(Y|\mathrm{do}(T), T \rightarrow Y)$. **Right:** When provided with the wrong causal information $T \leftarrow Y$, the model makes the prediction $Q_\theta(Y|\mathrm{do}(T), T \leftarrow Y) = P(Y|\mathrm{do}(T), T \leftarrow Y)$, which is the correct thing to do under the wrong assumption $T \leftarrow Y$. In this case, the causal effect is not identifiable and thus the observational data does not provide any indication about the causal direction.

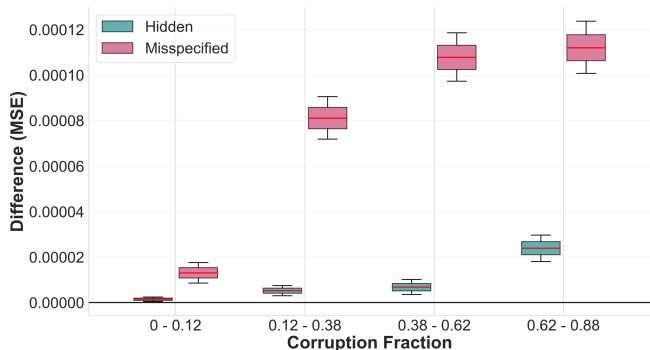

*Figure 20.* Hiding vs. misspecifying edges in the partially known ancestral matrix, evaluated on 100,000 samples from our complex mechanism prior. The baseline is a graph-conditioned model (GCN + Soft Attention) receiving the full ancestral matrix; we compare it against the same model when a fraction of entries is corrupted. *Hidden*: selected entries $T_{i,j} \in \{-1, 1\}$ are replaced by $T'_{i,j} = 0$ (marked as "unknown"). *Misspecified*: the same fraction of entries has its sign flipped, $T'_{i,j} = -T_{i,j}$. Misspecifying edges leads to a rapid and substantial performance drop, while hiding the same edges degrades performance only gradually. Practitioners who are uncertain about a causal relationship are therefore far better off leaving it unspecified than guessing—a key practical advantage of supporting partial ancestral matrices.

