# OpenReview forum: "Use What You Know: Causal Foundation Models with Partial Graphs"
_ICML.cc/2026/Conference — ICML 2026 regular_

### Official Review · Reviewer_Xf8P · 2026-03-03

**Soundness:** 3
**Presentation:** 3
**Significance:** 3
**Originality:** 3
**Overall Recommendation:** 4
**Confidence:** 3

**Summary:**

The authors aim to integrate background knowledge into Prior-Data-Fitted Causal Foundation Models (CFMs). The authors discuss multiple forms of background knowledge, such as direct and ancestral causal relations, and propose multiple methods to utilize them. The authors evaluate their methods on synthetic and semi-synthetic data.

**Compliance With Llm Reviewing Policy:**

Affirmed.

**Final Justification:**

The rebuttal addressed my main concern about interpreting the results so I have increased my score. In general, I believe the paper is sound and the contributions are useful for those who use foundation models for causal discovery.

**Key Questions For Authors:**

1. My main concern, as detailed above, is that it is not clear how and why the GCN architecture influences the results even when no ancestral information is provided. I would expect that when no information is provided, the models should be equivalent to regular CFMs. However, the results, especially in Figure 4, contradict this. Why does the GCN approach, even without ancestral information, improve results and why is this a sound experimental result?
2. How are the linear SCMs exactly generated with single layer perceptrons?

**Limitations:**

yes

**Strengths And Weaknesses:**

## Soundness

The paper mostly focuses on experimental results. My main concern with the experimental results is that I do not understand how encoding **zero** ancestral information (Hide Fraction of 1) in a GCN improves the results in Figure 4, especially for NLL. The authors state that this result

>“(...) indicates a consistent benefit of training models **with ancestral information** on the complex mechanisms, even when **no ancestral information** is provided.“

This sentence is especially confusing to me as I believe this should definitely not happen. If no ancestral information is provided, the methods should be equivalent to the baseline. I wonder whether the GCN simply provides a better architecture for the CFMs, regardless of ancestral information, in which case it is unfair to compare it to the baseline. Until the authors clarify why this result arises, I would not consider these results sound.

Similarly, I am also confused about why there is difference between “Partial” and Complete Ancestral Information methods for a Hide Fraction of 0 in Figure 10, especially for the $R^2$ results. Can the reviews elaborate why this is the case?

In general, I believe Section 5.2, the case of partial ancestral knowledge, should be discussed and experimented on more than Section 5.1, since full ancestral knowledge is an unlikely scenario in practice.

## Presentation

In general, the paper is clearly written and is easy to follow.

I am missing the description of the edge density for the simulated graphs, as I could not find this even in Appendix F.1. I am also confused about what it means that “the linear mechanisms are implemented as one-layer PyTorch ~ML~Ps” with default weight initialisation”. Does each dimension in the input correspond to a feature? In that case, with default initialisation, this would create a fully connected and cyclic graph since every feature has a direct effect on every other feature. I would appreciate it if the authors clarified how they generate the SCMs.

Section 5.1 should explicitly state that all ancestral or direct causal relations are provided to the models.

## Significance

The question that the authors aim to answer, what knowledge should be utilized in CFMs and how, is relevant and interesting. I believe this work can be helpful for future research and applications of CFMs.

## Originality

While the paper builds on existing work on CFMs, the proposed extensions on multiple ways to utilize background knowledge is original. Additionally, the authors develop a test-bed more suitable to train CFMs on.

### Smaller comments

I do wonder why the authors decided to use the notation $x \succ y$, instead of $x \prec y$, to denote that $x$ is an ancestor $y$, when in my experience, the latter is much more common especially when talking about the topological order, which is highly related to ancestry in causal DAGs (since $y$ “succeeds”, hence \succ, $x$ in the order). In fact, I wonder why this notation is introduced when another notation $x \leadsto y$ is introduced in the *preceding* sentence already for the same thing.

---

> ### Author Rebuttal · Authors · 2026-03-29
>
> Thank you for taking the time to review our paper. We are pleased to hear that you find that
>
> > I believe this work can be helpful for future research and applications of CFMs.
>
> ## Questions
>
> > My main concern […] is that it is not clear how and why the GCN architecture influences the results even when no ancestral information is provided.
>
> We thank the reviewer for raising this important point and for the opportunity to clarify our experimental setup.
>
> To study whether graph-conditioned models can amortise over varying levels of graph information, we compare them to two non-amortised baselines: one trained without conditioning and one trained with full conditioning. Therefore, we train three separate models with the same architecture (except for the GCN component), but **different training data**. More specifically, in this setup the first baseline is trained without any graph information; the second one with perfect graph information, and the actual model with **varying amounts** of graph information.
>
> In the limit of infinite data, model capacity, and training time, we expect: 1) The amortised model evaluated without graph information to match the non-conditioned baseline. 2) The amortised model evaluated with full information to match the fully conditioned model. Empirically, this behaviour holds for MSE and $R^2$ across both synthetic settings. The only deviation is the NLL improvement noted by the reviewer.
>
> We believe this effect is explained by training dynamics. Conditioning information makes many meta-learning tasks easier, particularly early in training. Exposure to these easier tasks may
> guide the model toward better regions of the loss landscape. In particular, attention biasing encourages the model to focus on causally relevant features, which can provide a useful inductive signal during training. The non-amortised baseline cannot benefit from this curriculum effect, even when no graph information is provided at test time.
>
> This is confirmed by our new experimental results, where we train an additional model that includes the GCN component but receives no graph information during training: https://anonymous.4open.science/r/4C77/GCN_NoGraph.pdf
>
> This model performs slightly **worse** (though not significantly) than the baseline without a GCN. This confirms that the gains of graph-conditioned models do not stem from architectural changes alone, but from the use of graph information during training.
>
> > How are the linear SCMs exactly generated with single layer perceptrons?
>
> Thank you for pointing this out. The SCMs themselves are not implemented as fully connected MLPs over all variables. Instead, the causal graph is sampled first, and the mechanisms are constructed respecting the parent structure of the DAG.
>
> Concretely, for each node we first sample its parent set from the DAG. A corresponding linear **mechanism** is then implemented as a single-layer perceptron that only takes these parents as input.
>
> We will extend the description of our prior accordingly in the Appendix of the paper.
>
> ## Further Concerns
>
> > Similarly, I am also confused about why there is difference between “Partial” and Complete Ancestral Information methods for a Hide Fraction of 0 […]
>
> The explanation is analogous to that of the first question above. These results correspond to different models trained under different conditioning regimes. While amortisation generally leads to similar performance to the specialised models, small differences are expected due to differences in training distributions.
>
> > Section 5.1 should explicitly state that all ancestral or direct causal relations are provided to the models.
>
> We will add a sentence making this very clear in the revised version of the manuscript.
>
> > In general, I believe Section 5.2, the case of partial ancestral knowledge, should be discussed and experimented on more than Section 5.1 [...]
>
> We agree that working with partially known structure is particularly important in practice. As fully known graphs are a standard assumption in parts of the causal inference literature [1], we use this setting as a controlled test-bed for our linear-Gaussian experiments where the effect of graph conditioning can be studied without additional uncertainty from missing structure.
>
> Motivated by your suggestion, we additionally evaluated the effect of conditioning on partial ancestral information on the DoPFN prior [2], obtaining results consistent with our other benchmarks:
> https://anonymous.4open.science/r/4C77/DoPFNPrior.pdf
>
> > I do wonder why the authors decided to use the notation […]
>
> This is a good suggestion. We will unify the notation and consistently use $x \leadsto y$ in the revised manuscript.
>
> ## References
>
> [1] Pearl, Causality. Cambridge University Press, 2009.
>
> [2] Robertson et al., Do-PFN: In-Context Learning for Causal Effect Estimation. NeurIPS 2025.

---

> > ### Author Rebuttal · Reviewer_Xf8P · 2026-04-01
> >
> > Thank you for your detailed rebuttal and for answering my concerns. I think generally I understand and agree with your explanation. However, I am still a bit confused about the exact difference between "GCN + Soft attention (ancestor)" and "Soft attention (ancestor)" for Figure 4 and it would be very helpful if you very explicitly stated the difference both here and in the paper. You say that the difference is explained by the fact that "Conditioning information makes many meta-learning tasks easier, particularly early in training." But what do you mean by "conditioning"? Isn't the "(ancestor)" component in "Soft attention (ancestor)" indicate a conditioning on ancestral information during training? Can you please very clearly state what the difference is in training between GCN + Soft attention (ancestor)" and "Soft attention (ancestor)"? In general, it would be much more clear if Figure 4 compared two "GCN + Soft attention"with different training information so the architectural differences are controlled for, even if the GCN component is, if I understand correctly, unused in one of them.

---

> > > ### Author Response · Authors · 2026-04-01
> > >
> > > Thank you for your questions! We are very happy to hear that you generally understand and agree with our explanation.
> > >
> > > We also thank you for the opportunity to clarify the exact difference between "GCN + Soft Attention (ancestor)" and "Soft Attention (ancestor)" in Figure 4.
> > >
> > > In our complex-mechanisms experiment, shown in Figure 4, we train and evaluate three models in total.
> > >
> > > - First, a baseline model (let's call it "model B") that operates exactly like existing Causal Foundation Models and does **not** have a GCN component or any sort of attention biasing. It is trained and evaluated without any graphical information.
> > >
> > > - Second, the Soft Attention (ancestor) model ("model SA"): this model has basically the same architecture as model B, but is trained and evaluated with partial graph information. The conditioning on graphical information is implemented via one additional bias term per attention head per layer (which leads to 24 additional parameters). The architecture of model SA is thus extremely similar to that of model B, but crucially, the training data include the partial graphs.
> > >
> > > - Third, the GCN + Soft Attention (ancestor) model ("model SA+GCN"): this model has the same architectural backbone as model SA, and is trained with the same training data as model SA, but crucially, has an **additional GCN component** to condition on the graph.
> > >
> > > Figure 4 shows the **difference** in performance of model SA relative to model B and the difference in the performance of model SA+GCN relative to model B.
> > >
> > > What we find here is:
> > >
> > > - Model SA outperforms the baseline in all metrics if sufficient graph information is present. We can see this by the confidence intervals not intersecting zero on the y-axis. Therefore, even for extremely similar architectures, graph-conditioning is beneficial across the entire hiding range.
> > >
> > > - Model SA+GCN often shows an even larger improvement over the baseline, indicating that the GCN component can provide further benefits.
> > >
> > > - Surprisingly, the overperformance of models SA and SA+GCN holds even for a hiding fraction of 1.0 at inference time. As mentioned in our rebuttal, this was a surprising finding for us too, which we would not expect in the limit of infinite data, compute, and capacity. Outside that limit, training dynamics differ between model B and models SA and SA+GCN—caused by the introduction of the additional conditional graphical information at training time—which is what we observe here.
> > >
> > > - In our new additional results: https://anonymous.4open.science/r/4C77/GCN_NoGraph.pdf we find that adding the GCN component to a regular CFM without graph-conditioning, i.e. the same training setup as for model B but still using the GCN component, performs slightly **worse** than the baseline without graph-conditioning. i.e. we already compare against the stronger baseline among the two architectural baselines.
> > >
> > > We would also like to point out that it is expected that the additional GCN component will not be beneficial if we train without graph conditioning. This is because the GCN can be seen as an additional encoder whose representation is used to modulate the representations of the main tabular transformer model via Adaptive Layer Normalisation. If the GCN does not receive any input during training, it is basically impossible that it also receives a meaningful training signal.
> > >
> > > We once again thank the reviewers for asking these clarifying questions. We will make sure to very precisely state in the revised version of the manuscript (a) what exactly the architectural differences between the models are and (b) what exactly they are trained on. Concretely, we will include a list, very similar to the one in this response, in the paper.
> > >
> > > > In general, it would be much more clear if Figure 4 compared two "GCN + Soft Attention" with different training information [...]
> > >
> > > We also created precisely the plot the reviewer asks for: the GCN + Soft Attention model's difference from the baseline that uses the same architecture but does not use graph-conditioning ("model B+GCN"): https://anonymous.4open.science/r/4C77/GCNNograph_vs_GCNGraph.pdf
> > >
> > > This plot shows exactly the same qualitative results as the one in our Fig. 4, albeit with minimally larger margins since the GCN + Soft Attention baseline is slightly worse than just the baseline B (https://anonymous.4open.science/r/4C77/GCN_NoGraph.pdf).
> > >
> > > We do not use this plot in the main body of our paper because in addition to assessing the effect of graph-conditioning over the baseline, our goal is to compare the two most promising graph-conditioning methods: just using Soft Attention and using Soft Attention together with the GCN. This makes more sense if both approaches are compared to a single baseline (model B), and we choose this baseline already as the stronger option among B and B+GCN.
> > >
> > > If this addresses the reviewers' concerns, we would greatly appreciate it if you could consider reflecting this in your updated rating of the paper.

---

### Official Review · Reviewer_wVk6 · 2026-03-04

**Soundness:** 3
**Presentation:** 2
**Significance:** 3
**Originality:** 4
**Overall Recommendation:** 4
**Confidence:** 4

**Summary:**

This paper considers the problem of incorporating domain knowledge in the form of (possibly partial) ancestral relationships between variables in prior-data fitted networks (PFNs). The prior data used in this work are generated from structural causal models (SCMs), assuming all relevant variables are observed. The work proposes a new architecture to incorporate the additional ancestral information given with observed data, and investigates different ways to condition on the additional information.

**Compliance With Llm Reviewing Policy:**

Affirmed.

**Final Justification:**

My main concerns have been addressed.

**Key Questions For Authors:**

Major questions:

1. I find the 2nd paragraph in Section 2.2 extremely confusing. The authors claim that Bayesian approach allows for weaker assumptions, in what sense? And what Bayesian approach is referred to here? From a causal identification perspective, Bayesian or any other approach does not automatically relax any identification assumptions.
2. Can the authors discuss and/or demonstrate how robust the proposed model is against misspecified or partially misspecified ancestral information?
3. Can the authors discuss and/or demonstrate how unobserved variables may affect the model’s performance?

Minor questions:

4. Grammatical mistake e.g., “Neural processes are models that meta-learn to mapping from datasets to a complex, usually intractable, posterior predictive of interest [...]”
5. Typo: “[...] while checkinf for all pairs”

**Limitations:**

I think limitations are not well addressed in the current paper. As mentioned above, there are at least two main limitations of the proposed model: 1) unobserved variables; 2) misspecified ancestral information.

**Strengths And Weaknesses:**

Strength:

- [Originality] Incorporating domain knowledge via (partial) ancestral information to causal PFNs is novel to my knowledge. The paper also investigates specific architectures to incorporate the given information.
- [Soundness] The argument and analysis presented look sound to me.
- [Significance] The paper aims to address the problem of incorporating additional information in a causal foundation model, which I believe is of interest in the field.

Weakness:

- [Presentation] The paper is overall well written, but there are some confusing paragraphs, typos, and grammatical mistakes. See Key Questions for authors.
- [Significance] This work assumes no hidden variables, which is usually not verifiable in real data and could easily be violated in reality. The paper did not discuss or demonstrate the consequences of this violation.

---

> ### Author Rebuttal · Authors · 2026-03-28
>
> Thank you for your positive and helpful review. We are very happy to hear that you assess the originality of our paper as excellent.
>
> ## Questions
>
> > 1. I find the 2nd paragraph in Section 2.2 extremely confusing. [...]
>
> Thank you for pointing this out. While Bayesian approaches to causality have a long history [1], and have been introduced in the context of causal foundation models by, e.g., [2] we will clarify this section.
>
> In “classical” Pearlian causality, one typically assumes a fixed, single SCM $\psi_0$ inferred from domain knowledge or data $\mathcal{D}$; causal inference is performed with respect to this fixed model $\psi_0$.
>
> In practice, however, a single SCM might not be identifiable, especially from observational data alone. Our Bayesian approach instead considers a posterior distribution $p(\psi \mid \mathcal{D})$ over multiple plausible SCMs. From this perspective, the classical approach corresponds to assuming a degenerate posterior $p(\psi \mid \mathcal{D}) = \delta_{\psi_0}$.
> While a fixed SCM defines interventional distributions via $p(y \mid do(t), x, \mathcal{D}) = p(y \mid do(t), x, \psi_0)$, the Bayesian approach yields
>
> $ p(y \mid do(t), x, \mathcal{D}) = \int p(y \mid do(t), x, \psi) p(\psi \mid \mathcal{D}) d\psi.\ (1).$
>
> We can thus weaken the assumption of a fixed SCM $\psi_0$ by replacing it with a posterior belief $p(\psi \mid \mathcal{D})$ over plausible SCMs induced via a prior $p(\psi)$. The effect of not precisely specifying the SCM is a potentially less concentrated posterior $p(y \mid do(t), x, \mathcal{D})$ because the uncertainty in our belief $p(\psi \mid \mathcal{D})$ needs to be propagated to our final prediction (Eq. 1). This Bayesian approach is subject to the same identifiability criteria/properties as non-Bayesian methods, but can additionally operate under weaker assumptions while accounting for the uncertainty this induces.
>
> We will clarify this explanation in the revision and refer to [2] for further background on the Bayesian framework in the context of causal foundation models.
>
> > 2. Can the authors discuss and/or demonstrate how robust the proposed model is against misspecified or partially misspecified ancestral information?
>
> First, if the causal structure is not identifiable from data, the given causal graph can completely determine the causal effect, which we show here: https://anonymous.4open.science/r/4C77/MisspecQual.pdf
>
> If this information is misspecified, the model's performance is expected to degrade quickly when more and more edges are misspecified; this is, however, equally expected in traditional methods [1].
>
> Furthermore, we quantitatively demonstrate the effect of misspecification on our model in an additional evaluation: https://anonymous.4open.science/r/4C77/MisspecExp.pdf
>
> Importantly, we find that **marking an edge as unknown is substantially less harmful than specifying it incorrectly**. This highlights an important practical and unique advantage of our method, as it utilises partially known ancestral matrices: practitioners need not commit to a fully specified DAG when some relationships are uncertain. Instead, what is unknown can be treated as such.
>
> In the revised paper, we will clarify that robustness to misspecification is limited in essentially the same way as for traditional causal approaches, while the ability to leave edges unspecified is a more specific practical benefit of our approach.
>
> > 3. Can the authors discuss and/or demonstrate how unobserved variables may affect the model’s performance?
>
> Under causal sufficiency, valid adjustment sets are guaranteed to exist (Corollary 3.2.6 in [3]), which is why we focus on this setting: it ensures that the model can in principle always find correct adjustment sets when sufficient graphical information is available. Without this assumption, causal effect identification is generally not possible without further assumptions (e.g. front-door settings), as discussed extensively in, for example, [3].
>
> While we think that focusing on the case where causal sufficiency holds is the most reasonable first step, it is, however, directly possible to train a model in the same way we do, while also considering unobserved variables.
>
>
> We will expand the discussion of this limitation in the Limitations section of our paper for the revised version of the manuscript and make sure to clarify this assumption in the Introduction and Methodology sections as well.
>
> ## References
>
> [1] Daniels, et al. "Bayesian nonparametrics for causal inference and missing data". Chapman and Hall/CRC, 2023.
>
> [2] Dhir, et al. "Estimating Interventional Distributions with Uncertain Causal Graphs through Meta-Learning." NeurIPS 2025.
>
> [3] Pearl, Causality. Cambridge University Press, 2009.

---

> > ### Author Rebuttal · Reviewer_wVk6 · 2026-04-02
> >
> > I thank the authors for their explanation and additional experiments. I keep my original score.

---

> > > ### Author Response · Authors · 2026-04-04
> > >
> > > We thank the reviewer for the careful consideration of our rebuttal and for confirming that the concerns have been fully resolved. We are very glad that the additional explanations and experiments helped clarify the questions regarding robustness, assumptions, and presentation.
> > >
> > > If you believe the clarifications and additional evidence strengthen the paper compared to your initial assessment, we would of course greatly appreciate reconsideration of the overall score.

---

### Official Review · Reviewer_v7k6 · 2026-03-08

**Soundness:** 3
**Presentation:** 3
**Significance:** 3
**Originality:** 3
**Overall Recommendation:** 4
**Confidence:** 4

**Summary:**

This paper introduces a method to incorporate structural domain knowledge into Causal Foundation Models (CFMs) by conditioning them on partially known ancestral matrices (PAMs). While existing CFMs based on Prior-Data-Fitted Networks implicitly marginalize over all plausible causal structures, which lead to unnecessarily conservative and imprecise predictions, the proposed approach allows practitioners to flexibly specify relationships as a known cause, a known non-cause, or an unknown relationship. To inject this knowledge into a transformer-based architecture, the authors evaluate several mechanisms and find that combining learnable soft attention biases with a Graph Convolutional Network (GCN) for feature modulation is the most effective strategy. Through an amortized training procedure, the resulting model can seamlessly adapt to varying amounts of available causal information, demonstrating improved predictive accuracy on both complex synthetic datasets generated from a custom causal prior and the semi-synthetic RealCause benchmark.

**Compliance With Llm Reviewing Policy:**

Affirmed.

**Final Justification:**

I appreciate the further explanation and additional evidence provided by the authors, which greatly addressed my concerns. I'm happy to finalize my positive assessment.

**Key Questions For Authors:**

1. Could you elaborate on how the model is expected to handle incorrect entries in the partially known ancestral matrix (PAM)?

2. Could you clarify the exact implementation details of the Soft Attention Bias ? Specifically, are the learnable scalars ($\beta_{anc}$ and $\beta_{non-anc}$) shared globally across the model, or are they parameterized per-layer or per-head?

3. Can you broaden the empirical evaluation beyond the current synthetic setups and the single RealCause benchmark? Testing on more diverse real-world datasets would help establish practical generality.

**Limitations:**

Yes

**Strengths And Weaknesses:**

**Strengths:**
1. The paper addresses a highly relevant limitation of current PFN-style causal foundation models: they typically cannot incorporate available domain knowledge, leading to unnecessarily conservative posterior predictive estimates.
2. Conditioning on partial ancestral information (PAM) is a reasonable and practically plausible choice, as it allows for specifying "known cause," "known non-cause," or "unknown" without requiring a full adjacency matrix.
3. The proposed mechanisms that injecting learnable attention biases from the PAM and modulating features via a GCN are well-motivated, simple, and integrate cleanly into existing transformer architectures.
4. The presentation of the paper is clear and well-organized, which is easy to follow.

**Weaknesses:**
1. While the authors constructed a custom, natively causal prior using both MLPs and XGBoost mechanisms to generate complex data , the experimental evidence remains relatively limited in breadth. Most results are on synthetic setups, with only one semi-synthetic benchmark (RealCause).
2. The authors rightly acknowledge the lack of comprehensive real-world causal benchmarks in their discussion , but this current empirical scope still falls short of justifying the strong "all-in-one causal foundation model" framing.
3. The RealCause evaluation is encouraging but small in scope, and the authors explicitly note that performance is suboptimal on datasets like PSID due to severe class imbalance prior to manual rebalancing.
4. The "systematic evaluation" of conditioning strategies is a bit narrower than implied. There are limited ablations isolating exactly what drives the gains (e.g., parameterization choices, or scaling behavior in the number of variables).

---

> ### Author Rebuttal · Authors · 2026-03-28
>
> Thank you for your thoughtful and positive review. We appreciate the reviewer’s assessment that
>
> > The paper addresses a highly relevant limitation of current PFN-style causal foundation models.
>
> ## Questions
>
> > 1. Could you elaborate on how the model is expected to handle incorrect entries in the partially known ancestral matrix (PAM)?
>
> First, if the causal structure is not identifiable from data, the given causal graph can completely determine the causal effect [1], which we illustrate for our case here: https://anonymous.4open.science/r/4C77/MisspecQual.pdf
>
> In the case this information is misspecified, the model's performance is expected to degrade quickly when more and more edges are incorrect; this is, however, equally expected in traditional methods.
>
> Furthermore, we quantitatively demonstrate the effect of misspecification on our model in an additional evaluation: https://anonymous.4open.science/r/4C77/MisspecExp.pdf
>
> Importantly, we find that **marking an edge as unknown is substantially less harmful than specifying it incorrectly**. This highlights an important practical and unique advantage of our method, as it utilises partially known ancestral matrices: practitioners need not commit to a fully specified DAG when some relationships are uncertain. Instead, what is unknown can be treated as such.
>
> In the revised paper, we will clarify that, in general, robustness to misspecification is limited in the same way as for traditional causal approaches, while our method provides a way to mitigate this issue by simply leaving edges a practitioner is unsure about unspecified.
>
> > 2.) Could you clarify the exact implementation details of the Soft Attention Bias ?
>
> The learnable scalars are parametrised per layer and per head, resulting in $n_{layers} \times n_{heads}$ bias terms ($4 \times 6 = 24$ in the case of our architecture).
>
> This adds only a negligible number of parameters, compared to having a single bias term, while allowing different layers and heads to specialise in using the available causal information to different degrees.
>
> We will state this more clearly in the revised manuscript and expand the description of the attention-bias mechanism in both the main paper and the Appendix, especially since we agree with the reviewer that this form of attention modification is both simple and integrates well with existing tabular foundation models.
>
> > 3. Can you broaden the empirical evaluation beyond the current synthetic setups and the single RealCause benchmark?
>
> We agree that further real-world dataset evaluations would help to further underpin our findings. However, the lack of extensive, meaningful real-world datasets is (unfortunately) one of the biggest issues in the field of causality in general and not specific to our work. We refer to, e.g., [2] for a recent discussion of this issue.
>
> This also affects related work on Causal Foundation Models equally, e.g.
>
> - [3] evaluate only on a single dataset from RealCause, whereas we use all 4 datasets with, in total 310 realisations. [3] also considers 10 synthetic datasets whereas we evaluate on 30,000 synthetic datasets from our complex-mechanisms prior.
>
> - [4] evaluate only on a single real-world dataset.
>
> - [5] use simple synthetic datasets and the same RealCause benchmark that we also use.
>
> Beyond this, we take a meaningful step toward stronger benchmarking by introducing a diverse complex-mechanisms prior and evaluating it extensively in the predictive setting (first paragraph of Sec. 5.3 and App. E.2). Specifically, we show that training a "TabPFN-style" [6] model on this prior achieves very competitive predictive performance compared to commonly used tabular models on small datasets. To the best of our knowledge, no other work considers this form of empirical evaluation of the synthetic data simulators in the field of causal effect estimation. We believe that this generic, robustly implemented, and empirically validated causal prior is an important contribution to the current state of benchmarking for causal foundation models.
>
> To further validate our findings, we additionally evaluate the effectiveness of graph-conditioning on the prior used in DoPFN [5]. The results reinforce our findings on the linear, complex, and semi-synthetic benchmarks: https://anonymous.4open.science/r/4C77/DoPFNPrior.pdf
>
> ## References
>
> [1] Pearl, Causality. Cambridge University Press, 2009.
>
> [2] Poinsot, et al. "Position: Causal Machine Learning Requires Rigorous Synthetic Experiments for Broader Adoption." ICML 2025.
>
> [3] Ma, et al. "Foundation Models for Causal Inference via Prior-data Fitted Networks." ICLR 2025.
>
> [4] Dhir, et al. "Estimating Interventional Distributions with Uncertain Causal Graphs through Meta-Learning." NeurIPS 2025.
>
> [5] Robertson, et al. "Do-PFN: In-Context Learning for Causal Effect Estimation." NeurIPS 2025.
>
> [6] Hollmann, et al. "TabPFN: A Transformer That Solves Small Tabular Classification Problems in a Second." ICLR 2023.

---

> > ### Author Rebuttal · Reviewer_v7k6 · 2026-04-01
> >
> > I really appreciate the rebuttal and the additional evidence. It was helpful and addressed some of my questions. In particular, the authors clarified the implementation details of the soft attention bias, and the additional experiments strengthen confidence that the main trend is not limited to their custom complex prior. I also appreciate the added discussion of misspecified PAMs.
> >
> > However, my main remaining concern is still the breadth of the empirical validation. The overall evidence is still dominated by synthetic or semi-synthetic settings, which the paper itself acknowledges as a current limitation. The additional rebuttal experiments improve confidence, but they do not fully resolve this broader concern, and addressing it would require a more substantial expansion and is beyond the scope of a short rebuttal. Therefore, I am keeping my score unchanged which is already positive.

---

> > > ### Author Response · Authors · 2026-04-02
> > >
> > > We are very happy that we could clarify implementation details and that we could improve your confidence in our experimental results from a score of 3 to a score of 4.
> > >
> > > We also agree with the reviewer that real-world evaluation of our method would be highly desirable; however, it represents a critical challenge for the field more broadly (as discussed, e.g., in [1–3]).
> > >
> > > As part of our ongoing work, we are interested in exploring how our framework could be adapted to more domain-specific settings where such data may gradually become available (for example in certain biomedical or scientific applications where partial causal structure and limited interventional data can be obtained).
> > >
> > > However, acquiring this data requires close collaboration with domain experts and potentially domain-specific modelling choices (e.g., adapting the prior and data-generating assumptions), which we believe is an important direction for follow-up work rather than something that can be addressed within the scope of this paper.
> > >
> > > In this work, we therefore deliberately focus on the methodological question of how causal foundation models can incorporate partial structural knowledge in a generic and well-controlled setting. Our goal is to help establish the methodological foundations needed for such models so that they can be reliably applied once suitable real-world evaluation settings become available.
> > >
> > > To still provide as realistic an evaluation as possible, we constructed an empirically validated prior and demonstrated strong performance on established semi-synthetic benchmarks. Understanding how such results translate to fully real-world causal settings, and what additional ingredients may be required, remains an interesting and important direction for future research.
> > >
> > > ## References
> > >
> > > [1] Neal et al. RealCause: Realistic Causal Inference Benchmarking. arXiv, 2020.
> > >
> > > [2] Curth et al. Really Doing Great at Estimating CATE? A Critical Look at ML Benchmarking Practices in Treatment Effect Estimation. NeurIPS, 2021.
> > >
> > > [3] Poinsot et al. Position: Causal Machine Learning Requires Rigorous Synthetic Experiments for Broader Adoption. ICML, 2025.

---

### Official Review · Reviewer_YLhn · 2026-03-17

**Soundness:** 3
**Presentation:** 3
**Significance:** 3
**Originality:** 3
**Overall Recommendation:** 4
**Confidence:** 4

**Summary:**

The work proposes an approach to incorporate available domain knowledge into the so-called "Causal Foundation Models" under the assumption that no hidden variables exist. Previous approaches such as CausalFM, CausalPFN, and Do-PFN were only able to take the observational dataset as their input (under different settings) to estimate interventional targets. This paper extends those approaches so they can also take the partial causal graphs between the observed variables. They test two different ways to take the graph information (local attention biasing and conditioning on the partial graph representations via graph convolution networks). Their approach is validated against baselines that do not take such extra partial graphs as input and showcase a better finite-sample performance in general.

**Compliance With Llm Reviewing Policy:**

Affirmed.

**Final Justification:**

I thank the authors for providing a detailed rebuttal and producing new results to compare against other relevant baselines. I believe the current methodology contributes significantly to the research on causal foundation models. Moreover, the authors seem to be willing to tone down the significance of the experimental results, so my concerns about the soundness would be resolved. I've raised my scores accordingly.

**Key Questions For Authors:**

Q1) For the synthetic experiments in Section 5.1, why was a linear-Gaussian dataset selected? Is this because under linear link functions with Gaussian noise, the causal direction is ambiguous? Could the authors elaborate?

Q2) Since no unobserved confounding is assumed in the datasets of Table1, can they provide the results of other amortized methods like CausalPFN and CausalFM under this setting?

**Limitations:**

The paper does not sufficiently discuss the specific limitation of causal sufficiency (no hidden variables).

**Strengths And Weaknesses:**

## Soundness and Significance

The work does not discuss the nuances behind identifiability and how their approach overcomes it. Specifically, the paper explicitly assumes "no hidden variables throughout the work" (Lines 126-127). Under no hidden variables, the causal effects can be identified through the backdoor adjustment formula. This is because a valid adjustment set will always exist. This is not to say that the current work does not provide value and new insights. Of course, finding a valid adjustment set in general is not identifiable from the observational data alone and can benefit from more information such as partial causal graphs.

However, the current writing does not capture such nuances and presents itself as a method that can work with any of such settings. Some imprecise statements have been made. For example, in lines 222-224: _"consistency depends on the available information $\mathcal{G}\_0$ [causal graphical information], not the prior the model was trained on."_ But the notion of identifiability is more general than causal graphs per se. E.g., in the instrumental variable causal graph, the local average treatment effect becomes identifiable if we assume monotonicity (no defiers), while without that assumption and the same causal graph, the effects will be only partially identified.

Given the above discussion, the practical significance for identifiable no-hidden-variable settings is less clear without stronger specialised baselines. If there is no unobserved confounding in the datasets of Table1, then methods like CausalPFN and CausalFM (backdoor) also become applicable. However, there is no comparison against them in Table 1. My suggestion is to include experiments where the knowledge of partial graph can help identify valid adjustment sets (in settings where without the graph it is impossible to identify a valid adjustment set) and compare to the relevant baselines that take the entire covariate set as a valid adjustment set (e.g., CausalFM) to showcase the benefit of using graphs.

The assumption in Proposition A1 is also too strong. It assumes that for all $x_i$ there is a path between $x_i$ and $t$, as well as $x_i$ and the outcome. Standard unconfoundedness does not require all variables $x_i$ to be common cause.

Moreover, the claim in the abstract, which state that "[we] find that injecting learnable biases into the attention mechanism is the most effective method to utilize full and partial causal information," or that it has identified _"the most efficient method to inject [graphical] information"_ (line 411-412) is both strong, and is in contradiction with Figure 4 where it says GCN+soft attention bias is better on the complex synthetic setting.

In terms of the experimental results, the work shows that including ancestral information often results in better performance compared to no such information. However, it is not clear that the results are statistically significant. For example, in Figure 4, the absolute values of the gains in MSE and $R^2$ are very small ($O(10^{-4}), O(10^{-3})$). Even though the trend is monotonic, the confidence intervals mostly overlap among different fraction of ancestral information. In the meantime, in Figure 10 (Appendix H2), we see that there is a small difference between the case that the model is trained with no graph conditioning, and the case that it was trained conditioning on the graphical knowledge, but it hides that information during test. The authors state in line 1357-1358 that the differences are small and fall within confidence intervals. The paper should apply the same cautious language for similar statement when comparing the results in Figure 4.

## Presentation
I found the paper writing clear and mostly reproducible. The only part that may need improvement is the description of the Structural Modulation (GCN) in lines 284-295. A more elaborated description of this approach would benefit the readability of the method.

Minor typos: "knonw" in line 105, "checkinf" in line 226.

## Originality

I think including the domain knowledge (either through partial graphs or other means) will play a huge role in the development of amortized causal methods. This work takes a first step towards this goal and thus is novel and important. Certain ideas used in the work, like introducing attention biases to capture the ancestral relationships is clever and have the potential to be used in the future practical versions of this line of work.

---

> ### Author Rebuttal · Authors · 2026-03-28
>
> Thank you for your very detailed review! We are happy to hear that you think that
>
> > including the domain knowledge (either through partial graphs or other means) will play a huge role in the development of causal methods
>
> ## Questions
>
> >Q1) [...] why was a linear-Gaussian dataset selected
>
> For this specific experiment, we use linear-Gaussian mechanisms (with internal standardisation) as this implies that causal directions are not identifiable from observations alone [1]. This makes the (partial) graph the only causal signal, isolating the effect of graph-conditioning.
>
> > Q2) [...] can they provide the results of other amortized methods like CausalPFN and under this setting?
>
> The model proposed in this paper is trained on a generic prior over causal structures, and conditions on graphical information at inference time, while CausalFM and CausalPFN utilise individual models trained for specific identification settings, e.g., assuming that all covariates form a backdoor adjustment set.
>
> We additionally evaluate CausalPFN and CausalFM on our complex-mechanisms prior: https://anonymous.4open.science/r/4C77/BDComplex.pdf
>
> Here, CausalPFN, which is only designed for the backdoor case, performs similarly to our more generic method when the backdoor criterion holds but its performance degrades substantially more strongly than that of our method when it is violated. Similarly, CausalFM's performance strongly degrades when the backdoor criterion does not hold.
>
> Following your suggestion, we compare to CausalPFN, CausalFM, and also DoPFN, on RealCause: https://anonymous.4open.science/r/4C77/RealCause.pdf
>
> Unsurprisingly, models without graph conditioning show different performance---this is due to differences in the prior. CausalPFN demonstrates the best performance because its training data-generating process closely mimics the way RealCause was simulated. Meanwhile, CausalFM, DoPFN, and our unconditioned model perform similarly. Once graph conditioning is introduced, our method outperforms CausalFM and DoPFN across almost all datasets.
>
> ## Limitations
>
> > The paper does not sufficiently discuss the specific limitation of causal sufficiency
>
> We focus on the case with no hidden variables as a first step, but we expect that our approach could be directly extended to this case by also training with latent variables.
>
> However, this is an important limitation of our approach, and we will make it clear in the introduction, Section 4, and the Limitation section.
>
> ## Concerns
>
> > The work does not discuss the nuances behind identifiability and how their approach overcomes it
>
> Without graph information, our method amortises causal discovery, adjustment selection, and effect estimation, similar to [2], which also discuss identifiability. Importantly, this Bayesian amortised approach allows to propagate the uncertainty in finding a valid adjustment set to the final predictions of a causal effect.
>
> Furthermore, in our case the prior over SCMs is highly diverse and essentially nonparametric, which means that asymptotic point identifiability is not guaranteed through the prior itself [3], but can be recovered by our approach of conditioning on appropriate graphical information (also see App. B).
>
> > However, it is not clear that the results are statistically significant.
>
> We expect the differences between fractions of hidden edges to be small as hiding 25% more edges, selected uniformly at random, might not correspond to a large change in available information for identifying the causal query. The performance gap between the fully graph-conditioned model and the baseline remains statistically significant. The modest effect sizes stem from two design choices: (1) we use an empirically validated prior without modifying it to, e.g., artificially amplify the strength of causal interventions; and (2) we consider graphs with up to 50 nodes, where longer paths reduce average causal effect magnitudes (see Fig. 11).
>
> We will revise our claims regarding the NLL differences between the non-graph-conditioned and amortised models (Fig. 10, App. H.2) and clarify the description of Fig. 4.
>
> ## Further Points
>
> - We will add a comment to Proposition A1 explaining that not assuming a path between $x_i$ and $t$ as well as $x_i$ and the outcome can equally be specified in a partially known ancestral matrix while implying unconfoundedness.
>
> - We will provide more details on our exact findings regarding the best graph-conditioning method in the abstract.
>
> - We will expand the structural modulation description.
>
> ## References
>
> [1] Ormaniec, et al. "Standardizing Structural Causal Models." ICLR 2025.
>
> [2] Dhir, et al. "Estimating Interventional Distributions with Uncertain Causal Graphs through Meta-Learning." NeurIPS 2025.
>
> [3] Ma, et al. "Foundation Models for Causal Inference via Prior-data Fitted Networks." ICLR 2025.

---

> > ### Author Rebuttal · Reviewer_YLhn · 2026-04-01
> >
> > I thank the authors for responding to my review and providing additional experiments. Some of my main concerns, however, remain unsolved.
> >
> > First, the draft and the rebuttal are still giving more credit to the empirical results that what's reported. For example, the rebuttal states that _"... but its [(CausalPFN)] performance degrades **substantially more strongly** than that of our method when it is violated. Similarly, CausalFM's performance **strongly degrades** when the backdoor criterion does not hold."_ The degradation for CausalFM is $0.0347 \pm 0.0139$ and for CausalPFN, it is $0.0377 \pm 0.110$, while proposed method's degradation $0.0169 \pm 0.0103$. The reported uncertainty is large relative to the mean differences, and no direct uncertainty estimate for pairwise differences is provided. Hence, claims like _substantially more strong_ are misleading.
> >
> > Second, the additional experiments are not yet sufficient for a fair comparison. The rebuttal states, _"We additionally evaluate CausalPFN and CausalFM on **our complex-mechanisms prior**: ..."_ In other words, the comparison is done on benchmarks that are generated from the same prior the proposed method is trained on. Although it is a useful experiment for controlled ablation of the conditioning mechanisms, it does not provide a robust advantage in more realistic out-of-prior settings.
> >
> > Third, for more realistic benchmarks like RealCause, when compared to more baselines that achieve better results, the explanation is not fully convincing. The rebuttal states that _"CausalPFN demonstrates the best performance because its training data-generating process closely mimics the way RealCause was simulated."_, but not concrete evidence is provided for this claim. At present, the more realistic benchmarks are the one where the baselines with no graph conditioning perform better. This weakens the general empirical claim that graph conditioning yields practical benefits over current causal foundation models (CFMs). The rebuttal still does not clarify under what realistic conditions partial graphical knowledge leads to clear gains beyond toy or same-prior settings.
> >
> > I believe the idea of conditioning a general CFM on partial ancestral information is interesting, but the current experimental evidence does not yet support the effectiveness of this approach.

---

> > > ### Author Response · Authors · 2026-04-04
> > >
> > > We thank the reviewer for their new response and address the remaining concerns below:
> > >
> > > > The reported uncertainty is large relative to the mean differences, and no direct uncertainty estimate for pairwise differences is provided.
> > >
> > > Indeed, our previous analysis did not directly answer whether the graph-conditioned models degrade less than CausalPFN/CausalFM when the backdoor criterion is violated. The way to correctly address this is to perform paired bootstrapping on the exact same set of results (instead of the independent uncertainty quantification in our earlier response).
> > >
> > > Following the reviewer's suggestion, we compute the [difference in MSE](https://anonymous.4open.science/r/4C77/BDComplex_ss.pdf) between our method and CausalPFN as well as CausalFM as baselines for each dataset and differentiate between the case when the backdoor criterion is valid and when it is violated. We stress that in contrast to CausalFM and CausalPFN, our method is still well-specified when the backdoor criterion is violated.
> > >
> > > - Left (CausalPFN baseline): The graph-conditioned models (Soft Attention, GCN+Soft Attention) show improvements over CausalPFN. Importantly, this improvement becomes more pronounced when the backdoor criterion is violated.
> > >
> > > - Right (CausalFM baseline): All models (CausalPFN, ours) show improvement vs. CausalFM, with an increase in mean MSE difference observed for our methods when the backdoor criterion is violated.
> > >
> > > We also repeat [this experiment at a larger scale](https://anonymous.4open.science/r/4C77/BDComplex_ls.pdf) (30,000 vs. 3,261 datasets), confirming that these findings hold with a high level of statistical significance:
> > >
> > > **Summary:** Our additional analyses show that, in our controlled benchmark, the graph-conditioned outperforms CausalFM and CausalPFN when the backdoor criterion holds, and the improvement is more pronounced when the criterion is invalid.
> > >
> > > While we reiterate that we will follow the reviewer's important suggestions concerning the specific claims of statistical significance in the revised version of the paper, we view the existing results, together with the new findings above, as targeted and unambiguous findings regarding the effectiveness of our graph-conditioning in causal foundation models.
> > >
> > > > Although it is a useful experiment for controlled ablation of the conditioning mechanisms, it does not provide a robust advantage in more realistic out-of-prior settings.
> > >
> > > We agree that this experiment alone does not establish superiority in all realistic out-of-prior settings. Its purpose is more targeted: to isolate whether graph conditioning itself helps when identification-relevant structural assumptions are violated.
> > >
> > > Regarding the connection between realistic and out-of-prior settings, we would like to point out that TabPFN-style models generally rely on broad priors intended to cover a wide range of plausible real-world data-generating processes. Following this philosophy, we design a complex SCM prior spanning neural networks, Gaussian processes, and tree-based mechanisms—and verify its practical relevance by empirically demonstrating generalisation to real data, indicated by strong predictive performance on real-world datasets (see Fig. 5 in manuscript).
> > >
> > > Because our primary goal is to study the conditioning mechanism itself, evaluation on controlled prior-generated data allows us to isolate this contribution without conflating it with generalisation effects to the specific semi-synthetic data-generating process of RealCause.
> > >
> > > > The rebuttal states that "CausalPFN demonstrates the best performance because its training data-generating process closely mimics the way RealCause was simulated.", but no concrete evidence is provided for this claim.
> > >
> > > We agree that our previous rebuttal did not sufficiently justify this statement: RealCause and CausalPFN are closely aligned not only in the unconfoundedness assumption, but also in the functional form of the treatment assignment mechanism and in how treatment heterogeneity is controlled.
> > >
> > > Both RealCause and CausalPFN use a DGP of the form:
> > >
> > > $X \sim P_{\text{tabular}}, \ \ T \sim P(T\mid X), \ \ Y_t \sim P(Y_t\mid X)$.
> > >
> > > Here $P_{\text{tabular}}$ denotes the empirical distribution of covariates from existing tabular datasets, $T$ is a binary treatment, and $Y_t$ are the potential outcomes. While the two approaches use different $P_{\text{tabular}}$ distributions, they share this structural formulation.
> > >
> > > Furthermore, both methods use a neural-network-based propensity model of the form
> > >
> > > $P(T\mid X) = \mathrm{Bernoulli}(\sigma(f(X)))$
> > >
> > > where $f$ is parametrised as an MLP. In RealCause, this MLP is fitted to data, whereas in CausalPFN it is sampled from the prior.
> > >
> > > Furthermore, RealCause explicitly controls treatment heterogeneity via linear interpolation between CATE and ATE, whereas CausalPFN, according to its authors, applies the same "light-weight post-processing inspired by RealCause."

---

### Decision · Program_Chairs · 2026-04-30

**Decision:**

Accept (regular)

**Comment:**

The paper addresses an important limitation of current causal foundation models: incorporating available domain knowledge. Conditioning on partial ancestral matrices is a novel and practically meaningful idea, and the proposed attention-bias and GCN mechanisms are well motivated. Reviewers found the paper technically sound, original, and generally well written. Experiments support the claim that partial causal information improves performance. The main limitations are the narrow empirical scope, the assumption of no hidden variables, and limited robustness analysis. After rebuttal, some concerns have been addressed while others remain. However, these concerns do not outweigh the contribution. I recommend acceptance.